# Errors of Airborne Bathymetry LiDAR Detection Caused by Ocean Waves and Dimension-Based Laser Incidence Correction

**Kai Guo [1], Qingquan Li [1,*], Qingzhou Mao [2]**, Chisheng Wang [1], Jiasong Zhu [1], Yanxiong Liu [3], Wenxue Xu [3], Dejin Zhang [1] and Anlei Wu [2]

1    MNR Key Laboratory for Geo-Environmental Monitoring of Great Bay Area & Guangdong Key Laboratory of Urban Informatics & Shenzhen Key Laboratory of Spatial Smart Sensing and Services, Shenzhen University, Shenzhen 518060, China; guokai@szu.edu.cn (K.G.); wangchisheng@szu.edu.cn (C.W.); zjsong@szu.edu.cn (J.Z.); djzhang@szu.edu.cn (D.Z.)
2    Institute of Aerospace Science and Technology, Wuhan University, Wuhan 430079, China; qzhmao@whu.edu.cn (Q.M.); wuanlei@whu.edu.cn (A.W.)
3    First Institute of Oceanography, MNR, Qingdao 266061, China; yxliu@fio.org.cn (Y.L.); xuwx@fio.org.cn (W.X.)
*    Correspondence: liqq@szu.edu.cn; Tel.: +86-153355294400

**Abstract:** Ocean waves are a vital environmental factor that affects the accuracy of airborne laser bathymetry (ALB) systems. As the regional water surface undulates with randomness, the laser propagation direction through the air–water surface will change and impact the underwater topographic result from the ALB system, especially for the small laser divergence system. However, the natural ocean surface changes rapidly over time, and uneven ocean surface point clouds from ALB scanning will cause an uncertain estimation of the laser propagation direction; therefore, a self-adaptive correction method based on the characteristics of the partial wave surface is key to improving the accuracy and applicability of the ALB system. In this paper, we focused on the issues of spatial position deviation caused by surface waves and position correction of the underwater laser footprint, and the dimension-based adaptive method is applied to attempt to correct the laser incidence angle. Simulation experiments and analysis of the actual measurement data from different ALB systems verified that the method can effectively suppress the influence of ocean waves. Furthermore, the inversion result of sea surface inclination changes is consistent with the surface wind wave reanalysis products. Based on the laser underwater propagation model in the strategy, we also quantitatively analyzed the influence of surface waves on laser bathymetry, which can guide the operation selection and data processing of the ALB system at specific water depths and under dynamic ocean conditions.

**Keywords:** airborne laser bathymetry; ocean wave; laser incidence correction; underwater topographic survey

## 1. Introduction

Airborne laser bathymetry (ALB) systems are suitable for simultaneously detecting land and shallow water seafloor data. Based on the transmission properties of visible light with wavelengths between 470 and 580 nm, ALB systems can obtain geographic spatial information and environmental characteristics from the land, ocean surface, water column and seafloor [1]. As aviation platforms are system carriers, ALB systems have more operational flexibility and higher detection efficiency than ship-borne sounding systems with the conventional terrain acquisition mode in areas where ships and personnel are difficult to access [2].

In view of the abovementioned advantages, the ALB system has the ability to conduct multidisciplinary research and has received extensive attention in shallow water areas that require integrated water and land detection. Examples include underwater topography surveys, seafloor sediment classification [2,3], water quality exploration [4–6] and geotechnical analyses [7]. Moreover, ALB technology provides a well-proven alternative

for ocean remote sensing communities in clear and shallow ocean areas [8]. Combining the application of the ALB system with other marine detection equipment could constitute an effective guarantee for enriching the detection information and realizing multilevel extraction in the shallow ocean area, especially for satellite-borne ocean remote sensing accuracy assessment and full-waveform multitarget detection [9]. Therefore, ALB technology has good prospects for achieving complementary advantages between different bathymetrical technologies.

However, the ocean is a complex physical system; the ocean surface is the interface between air and water which always maintains irregular undulations under the influence of currents, tides, winds and other dynamic factors. As shown in Figure 1, the wave height and wavelength of the point cloud of the water surface undulations obtained by the ALB system are not constant. This uncertainty of the water surface is an important factor reducing the accuracy of underwater target detection, which is reflected in the energy loss of the echo signal and echo waveform broadening [10,11]. In addition, changes in the propagation direction and velocity of the laser beam as it passes through air and water affect the detection accuracy of the systems [12,13]. The maximum refraction error from water undulations is between 1% and 2% of the depth of the area [14,15].

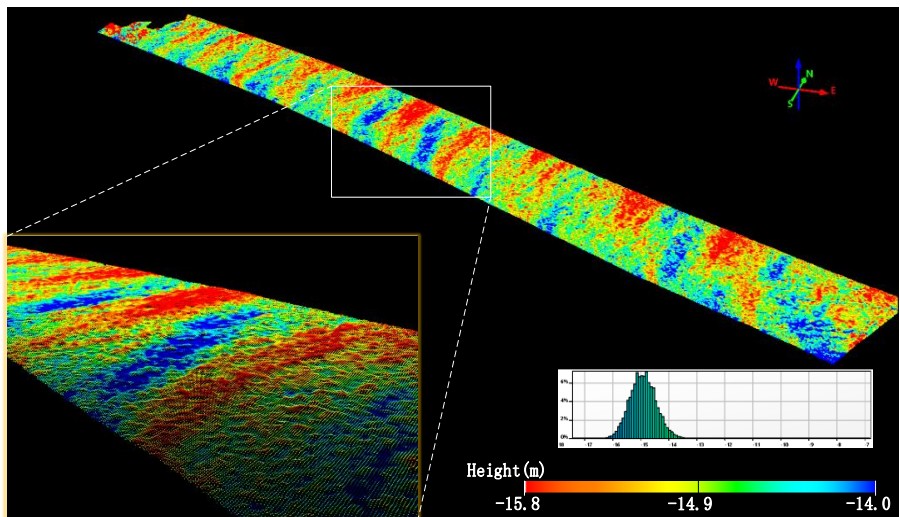

**Figure 1.** Water surface undulations represented by ALB point cloud.

In early research, the effects of refraction in dynamic water were mainly studied by stereophotogrammetry. Through the radial correction of image coordinates, the influence of water surface refraction deviation and beam radial error can be weakened [16–18]. On this basis, Maas and Mulsow further expanded the radial correction method to make it meet the application requirements of close-range photogrammetry when addressing nonplanar interfaces. However, this method does not perform well in the spatiotemporal treatment of wave surfaces [19,20]. Steinvall et al. [14,15] studied the characteristics of water surface undulations through the surface laser footprints obtained by the helicopter-borne light detection and ranging (LiDAR) system called FLASH and reduced the influence of the uncertainty of the water surface by simulation. It should be noted that the laser divergence angle of the FLASH system is approximately 2~10 mrad and may even exceed the wavelength range of water waves under working altitude and scanning angle conditions. However, the spot coverage area of the FLASH system is larger than that of ALB systems with a small divergence angle, while small-footprint LiDAR systems are more sensitive to water surface undulations [21]. Mandlburger et al. [22] used near-infrared (NIR, $\lambda$ = 1064 nm) and green ($\lambda$ = 532 nm) water surface echo signals to establish a digital water surface model with a regular grid structure. However, the large number of uneven echo signals around the target may have limited the interpolation accuracy of this water surface model. In addition, the Rigel company re-estimated the angle of incidence of

the water surface by establishing a water surface triangulation. It has been said that this method is applied to the data processing software RIHYDRO, but the details have not been disclosed [23].

Li [24] conducted a statistical study on the water surface slope caused by wind and waves and summarized the changes in the ALB system detection errors caused by wave surface inclination under different wind speed conditions. Huang et al. [25] proposed three alternative wave correction schemes based on the spatial geometric structure of laser propagation and deduced a rigorous formula for position calculation based on the full consideration of the dynamic effects of the ocean and the attitude of the carrier. Hu et al. [26] used aircraft altitude acceleration to estimate the ocean surface state. The method can meet the requirements of ALB system development to a certain extent. However, it does not consider the influence of surface undulations in the laser propagation process and is not suitable for water surface wave correction based on the raw detection results of an ALB system. Yang et al. [12] extracted the sea surface slope by using raw point clouds to fit the ocean surface based on the least square criterion and wave spectra and conducted laser refraction correction, which influences the position of the footprint. The modeling process takes the ocean surface point cloud in a large area as the basis of instantaneous wave surface fitting, and the model is too complicated, which affects its practicality. Due to the high complexity of the simulation of the instantaneous undulation surface, the precision may be affected by the raw point cloud density and noise related to the data acquisition process [27].

Based on the above analysis, correcting the incident laser direction by simulating the surface of ocean waves is feasible. However, limited by the Shannon–Nyquist theorem, sea surface point clouds with uneven temporal and spatial distributions adversely affect the accuracy of surface simulations [28–30]. When there are no other synchronous observation data at the laser scanning position, constructing a surface analysis method that can realize self-adaptive correction for the direction of laser incidence on the ocean surface, which is the main purpose of this paper, enables us to study the correction strategy around the following issues:

**(1)   Lower demand for ocean environmental parameters and system scanning mode.**

The demand for real-time sea conditions and environmental parameters is a vital reason for irregular water surface modeling, while the points obtained by ALB systems with different scanning modes are generally uneven. The method of region segmentation to refine the surface simulation will inevitably affect the accuracy of the data processing and increase the amount of calculation. Therefore, as an effective strategy for laser incidence correction, the robustness of the method for external environmental changes and system scanning methods should be increased and is of great value for improving the applicability of the system.

**(2)   Estimation of laser incident direction based on morphological characteristics of local water surface points.**

The deviation of the laser incident direction is related to the local inclination of the fluctuating water surface. If the incident surface that meets the special continuity characteristics could be directly identified based on the footprints near the incident position, the complexity of the surface modeling process will be reduced, and the accuracy of laser incidence correction could be improved effectively.

**(3)   Self-adaptive selection of point cloud neighborhoods for irregular incident surfaces.**

When the ALB system is flying and scanning, the water surface waves randomly and rapidly at the same time. Therefore, the initial water surface point cloud in the scanning path cannot accurately reflect the instantaneous state. In addition, as the scanning footprints are generally uneven [31–35], the calculation based on a fixed radius neighborhood will adversely affect the stability of the correction result. Realizing self-adaptive selection of the incident surface neighborhood can improve the practicability of the method.

In this paper, we study the method for correcting the laser incidence angle on undulating ocean surfaces and apply a self-adaptive correction model based on the raw point cloud dimension to improve the applicability of ALB systems. The simulated undulation surface data and the measured data are used to verify and analyze the accuracy and practicability of the correction model. We use measured data with two scanning modes from distinct experimental areas in the South China Sea. On this basis, the effect of surface waves on the depth detection capability of an ALB system is analyzed with a quantitative approach, which provides some effective references for ALB data acquisition.

## 2. Materials and Methods

To avoid high complexity in the online calculation of ALB data, many commercial systems generally do not provide a wave correction function for underwater detection in the preliminary processing of received signals. According to the principle of airborne LiDAR bathymetry, when the system scans the target area, it can synchronously acquire the echo information of the water surface, water column and seafloor from the full-waveform data. The water surface point cloud corresponding to the echo position of the underwater target can be obtained by preliminary calculation [36]. As the ALB system performs area detection in a scanning manner, the undulations of water surface point clouds affected by surface waves have obvious temporal and spatial differences [37]. Considering the high frequency of airborne LiDAR scanning, when the distance between each laser shot is less than half the length of the surface wave, a limited range of the surface undulations can be reflected in the corresponding point clouds. By estimating the surface slopes at the position of the incident laser, we can calculate the propagation direction of the center of the laser beam and then correct the spatial position of underwater targets influenced by continuous waves.

The laser incidence correction method suggested in this section mainly focuses on the estimation of the laser incidence normal vector and the propagation correction between the different media. Figure 2 shows a flow chart of the incidence correction process. First, based on the original ocean point cloud obtained by ALB system scanning, the surface and seafloor point clouds are divided before incidence correction processing. An index involving the propagation distance between the laser footprints calculated from the same echo signal is established, which is useful for further processing. In addition, multiscale denoising is performed on the point cloud of the water surface undulations to suppress the adverse effects of the noise acquired during the scanning process on the position of the water surface point cloud. On the basis of the above processing, due to the temporal and spatial limitations of the point cloud in the area during the scanning process of the ALB system, we use the dimension-based neighborhood radius selection method and calculate the laser propagation vector to adjust the spatial position of the underwater laser footprints.

### 2.1. Surface Point Cloud Multiscale Denoising

Generally, there are many useless laser points above the water surface, such as isolated points, very high points and very low points. These useless points usually come from the signal echoes of aerial targets with strong scattering or objects floating on the water surface during ALB system scanning. In addition, when the blue-green laser irradiates the complex and dynamic water surface, the echo component is often broadened and has apparent uncertainty due to the interference of the surface wave environment and penetration, which leads to a ranging error in the detection of the water surface.

Therefore, it is necessary to filter the water surface point clouds to reduce the interference of environmental deviations on the water surface point cloud before analyzing and processing the water surface obtained by the ALB system. The waves on the sea surface are mainly derived from the superimposition of swells and wind waves, which are affected by environmental factors such as wind speed, wind direction, ocean currents, tides and surface undulations; therefore, surface elevation fluctuations often show directional and irregular characteristics. In this paper, we use the wavelet multiscale decomposition method

to decompose the elevation components of the point cloud into different directions [38], assign the interference uncertainty to each direction wavelet coefficient component [39] and then filter the point clouds by the soft threshold method, thereby suppressing the deviation between the raw point clouds and the corresponding water surface undulations, which are the main cause of the noise.

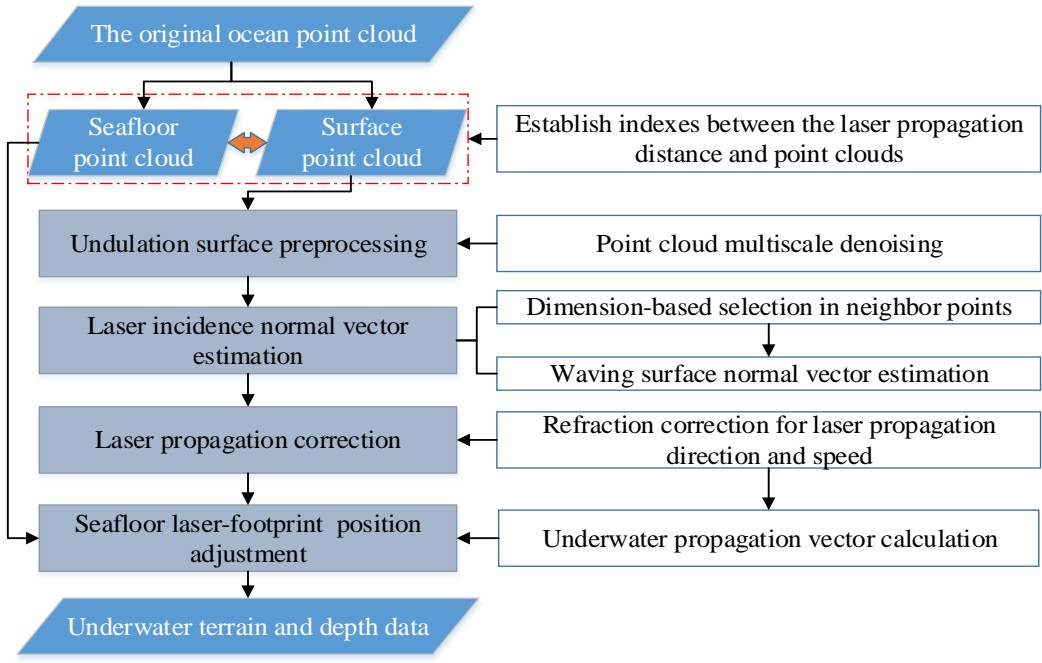

**Figure 2.** Flow chart of laser incidence correction for ocean surface undulations.

Assuming that the wavelet coefficients of the high-frequency components in the *j*-layer of the point cloud after wavelet transformation are $d^j_{mn}$, *m* and *n* are the corresponding row order and column order, respectively, of the sea surface point cloud, which is interpolated at equal intervals. The method to calculate the high-frequency wavelet coefficient component can be expressed as follows:

$$\tilde{d}^j_{mn} = \begin{cases} d^j_{mn} - \tilde{\varepsilon}_j, & d^j_{mn} \geq \tilde{\varepsilon}_j \\ 0, & |d^j_{mn}| < \tilde{\varepsilon}_j \\ d^j_{mn} + \tilde{\varepsilon}_j, & d^j_{mn} \leq -\tilde{\varepsilon}_j \end{cases} \tag{1}$$

$\tilde{\varepsilon}_j$ is the threshold of the corresponding two-dimensional high-frequency component. The wavelet coefficients of the effective signal must be larger than the wavelet coefficients of the noise, which have a small amplitude related to dispersive energy. Let the wavelet coefficients below the threshold become zero; then, the noise in the signal can be effectively suppressed. We estimate $\tilde{\varepsilon}_j$ as follows:

$$\tilde{\varepsilon}_j = \frac{\sum\limits_{m=1}^{N^j_m} \sum\limits_{n=1}^{N^j_n} |d^j_{mn}|}{0.6745 \times (N^j_m \times N^j_n)} \sqrt{2 \log(N^j_m \times N^j_n)} \tag{2}$$

Figure 3 shows the effect of wavelet multiscale decomposition of the surface point clouds obtained by the ALB system with db4 as the wavelet basis, and a 1-layer multiresolution analysis is performed. In the figure, the directional characteristics of the elevation distribution in the target area can be clearly observed. Since the wind direction is mainly northwest–southeast, the low-frequency component of the elevation changes mainly exists

in the corresponding direction (Figure 3a), while the high-frequency components that contain environmental noise are relatively weaker (Figure 3d). The high-frequency components are mainly distributed in other directions (Figure 3c,b).

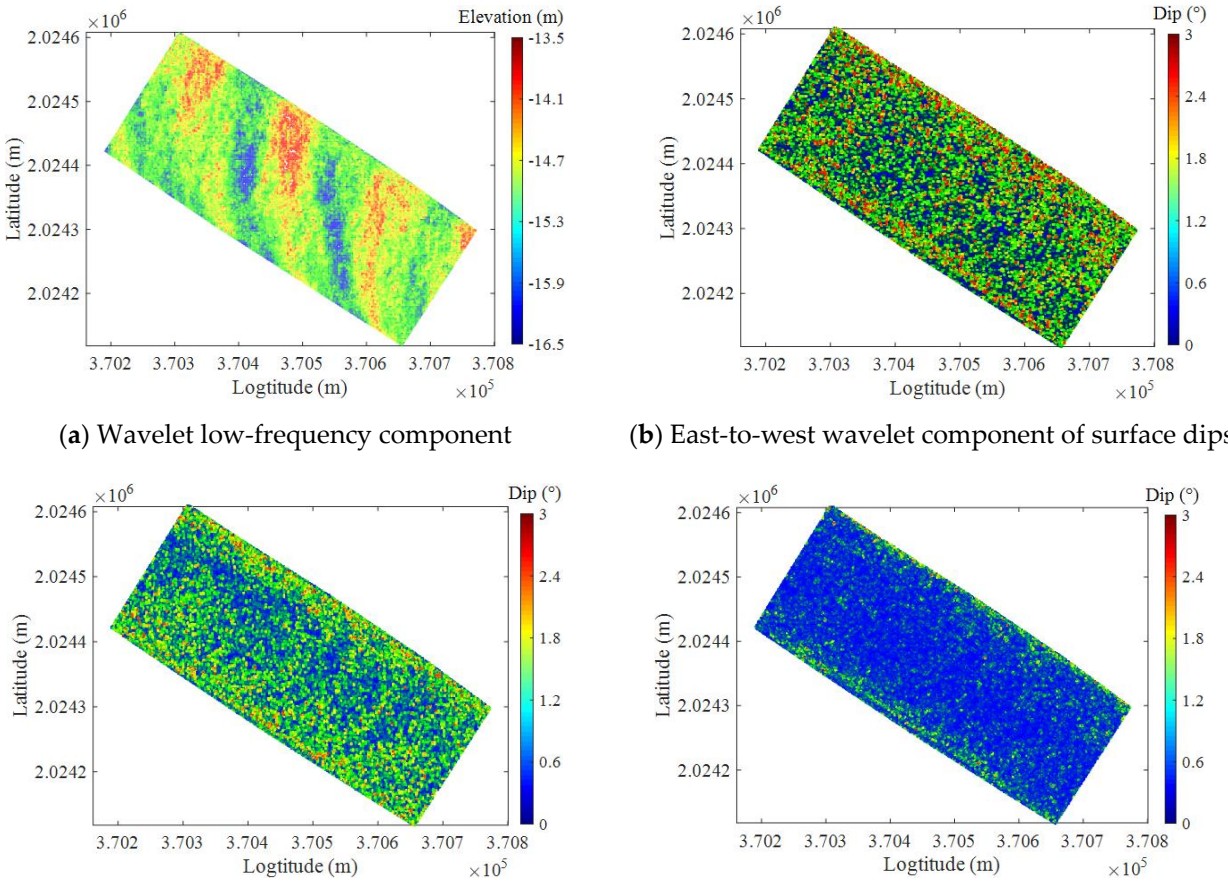

(**a**) Wavelet low-frequency component

(**b**) East-to-west wavelet component of surface dips

(**c**) North to south wavelet component of surface dips

(**d**) Skew wavelet component of surface dips

**Figure 3.** Multiscale decomposition (1-layer) of surface elevations and dips.

Wavelet soft threshold denoising is mainly carried out for the high-frequency components, and the filtered result can be restored to the water surface point cloud through the inverse wavelet transform.

### 2.2. Estimation of the Incidence Direction

The angle of incidence of the laser beam on the water surface is the critical parameter for calculating the three-dimensional position of the underwater laser footprint. There is a significant difference in the angle of incidence of the laser between a calm interface and an undulating surface [21]. The calm sea surface in Figure 4 is $S_3$, and its normal vector $\overrightarrow{n}$ is vertically upward. In the absence of surface undulations, the laser propagation direction in air is represented by $\overrightarrow{L}$, the normal vector $\overrightarrow{n}$ is coplanar within the plane $S_1$ and $\angle\alpha_1$ is the ideal laser incidence angle. As shown in Figure 4, there is actually an angle $\angle\alpha_3$ between the real laser incident plane $S_4$ and the ideal calm surface $S_3$, and this angle is the slope of the sea surface. The angle between the vector $\overrightarrow{v}$, which is normal to the plane $S_4$, and the vertical direction $\overrightarrow{n}$ is equal in numerical value to $\angle\alpha_3$. $\angle\alpha_2$ is the angle between the direction of the laser path in air and the normal vector $\overrightarrow{v}$.

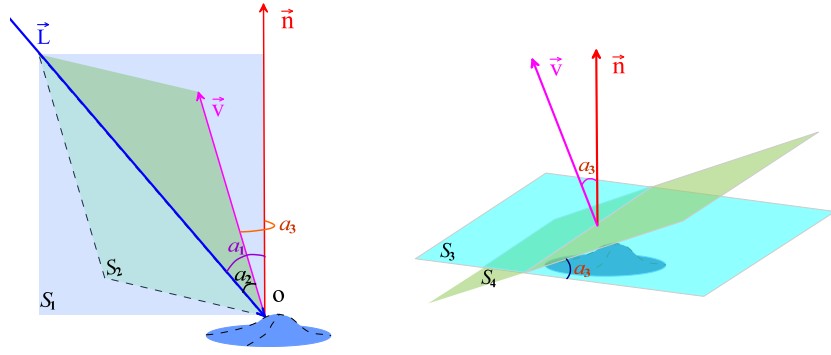

(**a**) Change in laser incidence direction          (**b**) Incident normal vector

**Figure 4.** Wave inclination and the change in the laser incident normal vector.

To re-estimate the laser incident angle, it is first necessary to calculate the deviation between the incident normal vector of the wave surface and the normal vector of the calm sea surface. This deviation can be decomposed into the slope and the aspect of the wave surface, where the aspect is the azimuth of the projection of $\vec{v}$ on the plane $S_3$.

Since the sounding system can quickly obtain laser point clouds through high-speed scanning, the laser footprints within the scanning coverage area have a certain distribution density. Based on spatial analysis of the laser points at the incident position and its neighboring points, a section of the local wave surface can be constructed. The deviation between the incident normal $\vec{v}$ and the vertical direction $\vec{n}$ can be calculated as the correction value for the laser beam incident angle. Figure 5 is a schematic diagram of the incident normal vector estimated according to the local characteristics of the laser foot point on the sea surface. The difference in point cloud normal vector directions in the area reflects the undulation of the local sea surface to a certain extent.

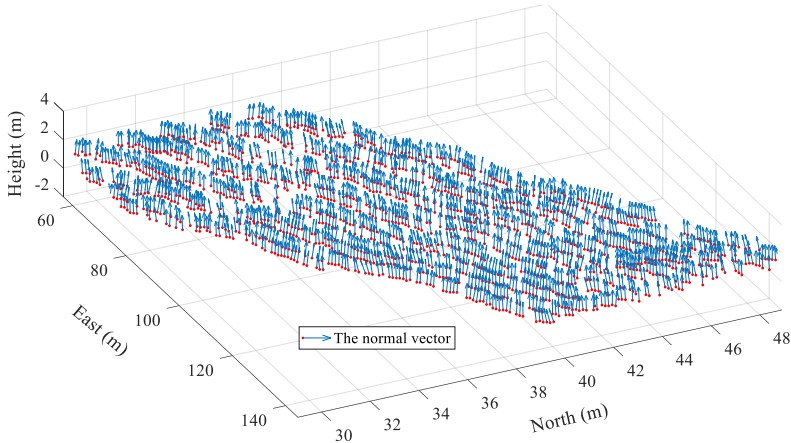

**Figure 5.** Schematic diagram of the normal vector of the laser incident on the sea surface.

The incident angle compensation corresponding to the foot points of the sea surface is the key to correcting the underwater laser reflection position. Therefore, the accuracy of sea surface wave correction is related to the estimation of the normal vector of the corresponding local sea surface.

### 2.2.1. Surface Normal Vector Estimation

Due to atmospheric scattering near the water surface and the uncertainty of the surface waves, the preliminary point clouds of the water surface obtained by the ALB system are not on a very smooth surface. Therefore, using the incident laser beam position and its neighboring points to simulate a local tangent surface that conforms to the least square

theory will effectively reflect undulation characteristics. The local simulation plane can be expressed as follows:

$$M(\vec{n}, l) = \text{argmin} \sum_{i=1}^{k} (\vec{n} \cdot p_i - l) \tag{3}$$

where $\vec{n}$ is the normal vector and $l$ is the distance between plane $M$ and the origin of the coordinate system.

To abate the adverse effect from the position error of the laser points, the gravity center position of the neighboring points is estimated and regarded as the tangent position of the corresponding reflection plane. According to principal component analysis (PCA), surface normal vector estimation is the process of analyzing the eigenvector corresponding to the minimum eigenvalue of the covariance matrix [40]. The covariance matrix is

$$C = \frac{1}{k} \sum_{i=1}^{k} (p_i - \overline{p})(p_i - \overline{p})^T \tag{4}$$

$$\overline{p} = \frac{1}{k} \sum_{i=1}^{k} p_i \tag{5}$$

$$C \cdot \vec{n}_j = \lambda_j \cdot \vec{n}_j \tag{6}$$

where $p_i$ is the position of neighbor points at a finite distance from the gravity center, $k$ is the number of neighbor points, $\lambda_j$ is the eigenvalue of the covariance matrix and $\vec{n}_j$ is the corresponding eigenvector. The eigenvector corresponding to the minimum eigenvalue is taken as the normal vector corresponding to the point cloud.

### 2.2.2. Dimension-Based Neighborhood Points Selection

The estimation of the normal vector of the wave surface lies in the selection of the neighbor points around the incident position. Under a fixed scanning frequency, the point cloud density often changes due to the scanning mode, the spatial distance between the object and scanner, noise or the flight speed. Certainly, it is necessary to set the neighborhood size with reference to the density of the water surface points to accurately analyze the effect of water undulations on the propagation direction of the laser beam, but this is very difficult in an operation with a single neighborhood radius.

Figure 6 shows the sea surface point clouds and the models of the surface correction values for the laser incidence angle. The distributions of the correction values calculated by the PCA method with different fixed neighborhood radii are displayed in Figure 6. Before we use a fixed neighborhood radius to correct the angle of incidence, resampling and interpolation are implemented on uneven surface point clouds.

In Figure 6, we can see that the deviations in the angle of incidence at the crests and troughs are small, while the deviations in the transition areas between the crests and troughs are larger. This indicates that the correction value for the laser incident angle is correlated with the degree of local surface inclination, which has a significant relation to the size of the normal vector estimation neighborhood.

When the neighborhood selection radius used for normal vector estimation is small, details of sea surface undulations can be expressed, but as the neighborhood radius gradually increases, some details of the surface topography in the area will also be lost gradually. Therefore, a reasonable neighborhood selection method for estimating the local normal vector is needed, which would have the benefit of reflecting the true state of water surface undulations and improving the estimation accuracy of the laser beam propagation process.

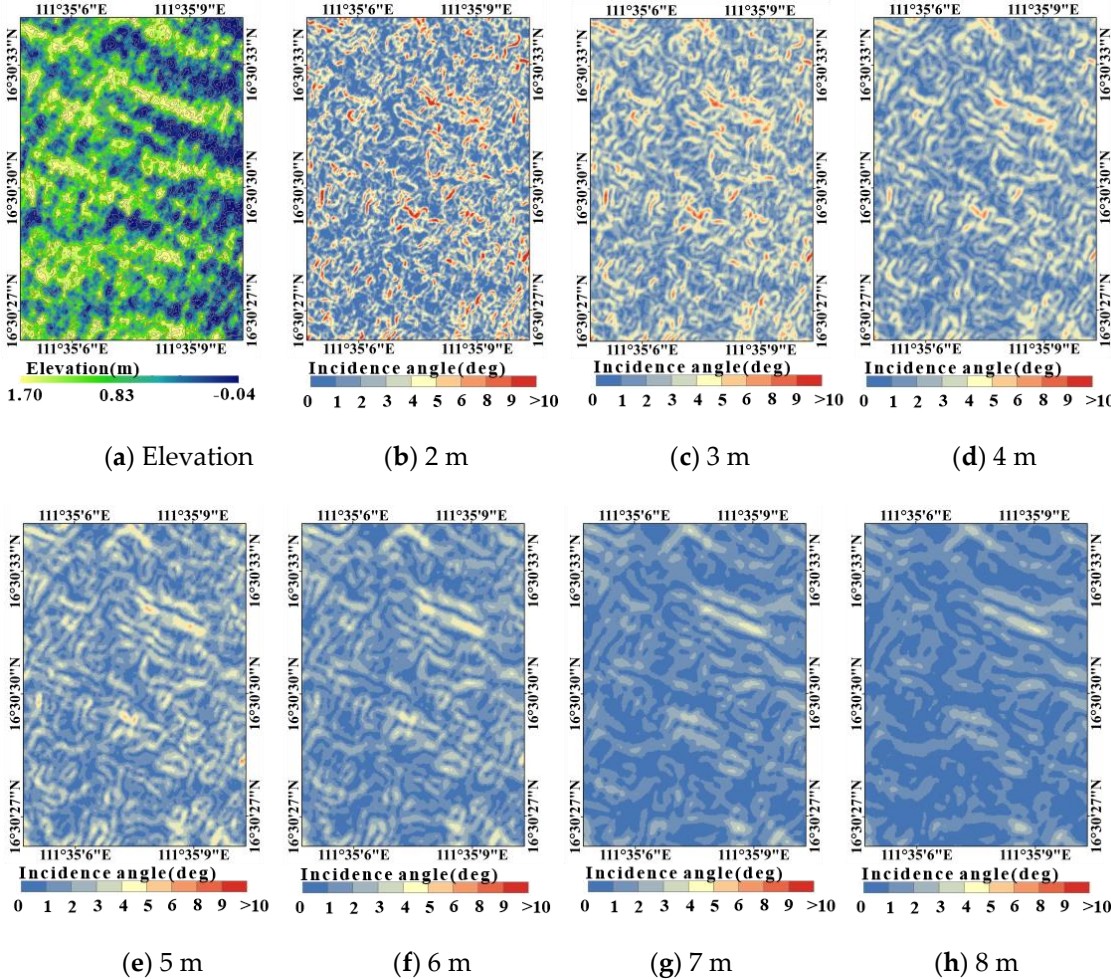

**Figure 6.** Incidence angle distribution on the ocean surface undulations ((**a**). Point cloud of the water surface, (**b**–**h**). Incidence angle distribution with different estimation radii.).

Therefore, we introduce a spatial feature dimensionality based on the neighborhood self-adaptive selection method to determine the optimal neighborhood for normal vector estimation [41]. By adaptively determining the optimal spatial neighborhood corresponding to the incident position, many problems such as neighborhood selection and complex surface function determination caused by local water surface fitting can be avoided. This method can significantly overcome the effects of the uneven density of point clouds and random noise. Finally, the local normal vector can be accurately calculated. The spatial feature dimension-based neighborhood adaptive method is described as follows:

(1) Initial parameter setting and initialization. The key to the neighborhood adaptive method lies in changing the neighbor point search radius, so some parameters need to be initialized first, which include the initial radius (minimum radius) $r_0$, maximum search radius $r_{max}$ and increment of the radius $\Delta_r$. The maximum search radius $r_{max}$ should not exceed the maximum distance between adjacent laser feet in the target point cloud, and the minimum search radius should be greater than the minimum distance between adjacent laser feet in the area. $O_{ls}$ is the laser beam incidence position on the water surface, and the laser points in the neighborhood are $\{|p_i - O_{ls}| \leq r_0 + ir_\varepsilon, \ ir_\varepsilon \leq r_{max}\}$, where $i = n - 1$, and $n$ represents the number of superpositions.

(2) Principal component analysis and feature dimensionality calculation. As the radius of the selected neighborhood points gradually increases from $r_0$, to $r_{max}$, PCA is performed on the data for each radius. The eigenvalues $\lambda_1$, $\lambda_2$ and $\lambda_3$ are calculated

according to Equation (6), where $\lambda_1 \geq \lambda_2 \geq \lambda_3$. Based on the eigenvalues, the values of feature dimensionality are calculated as follows:

$$a_{1D} = \frac{\sqrt{\lambda_1} - \sqrt{\lambda_2}}{\sqrt{\lambda_1}}, a_{2D} = \frac{\sqrt{\lambda_2} - \sqrt{\lambda_3}}{\sqrt{\lambda_1}}, a_{3D} = \frac{\sqrt{\lambda_3}}{\sqrt{\lambda_1}} \tag{7}$$

(3)  Calculation of the entropy of information and determination of the optimal neighbor points. As the neighborhood search radius continuously expands according to the radius step increment, the entropy of different neighborhood radius features is calculated, and the neighborhood radius $r_C$ corresponding to the minimum entropy $E_f$ can be selected as the optimal neighborhood by equation.

$$E_f = -a_{1D} \ln(a_{1D}) - a_{2D} \ln(a_{2D}) - a_{3D} \ln(a_{3D}) \tag{8}$$

### 2.2.3. Refraction Correction

When the laser propagates from the air to below the water surface, the laser is affected by refraction at the air–water interface, and the propagation direction of the laser changes from $\overrightarrow{OA}$ to $\overrightarrow{OA'}$, as shown in Figure 7. In addition, the speed of the laser in water is also different from that in air. Consequently, the refraction process must be considered in the calculation of the positions of the underwater laser points. The following will introduce the method of underwater point refraction correction based on the vector calculation.

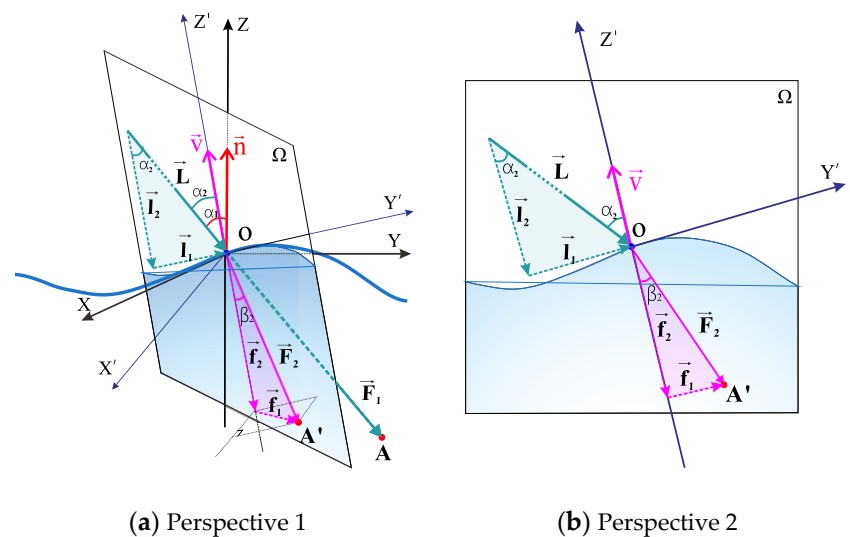

(**a**) Perspective 1          (**b**) Perspective 2

**Figure 7.** Laser refraction on a water surface undulation.

According to Snell's law, the effect of water surface refraction on the calculation of laser beam propagation is manifested in the direction and velocity of the laser.

$$\frac{\sin \alpha}{\sin \beta} = \frac{n_1}{n_2} = n_w \tag{9}$$

$$\frac{c}{v} = \frac{n_1}{n_2} = n_w \tag{10}$$

$\alpha$ and $\beta$ in Figure 7 are the incidence angle and refraction angle, respectively, $n_1$ is the refractive index in the water body and $n_2$ is the refractive index in the atmosphere. $v$ and $c$ are the velocities in the water and atmosphere, respectively.

The effect of changes in the refraction index on the calculation of seafloor coordinates in inhomogeneous media is generally less than the effect of changes in the incident direction and speed. To simplify the analysis, the small spatial variations in $n_1$ and $n_2$ are not

considered in this section. The time interval between the surface reflection and seafloor reflection is obtained by waveform processing. Let the propagation distance in the water without refraction correction be $|\vec{F_1}|$, and let $|\vec{F_2}|$ be the corresponding corrected distance. According to Figure 7, their formulas are

$$|\vec{F_1}| = |\vec{OA}| = ct \tag{11}$$

$$|\vec{F_2}| = |\vec{OA'}| = vt \tag{12}$$

The relationship between them can be expressed as follows:

$$|\vec{F_1}| = n_w |\vec{F_2}| \tag{13}$$

The incidence angle can be calculated as the angle between the normal vector and the center of the laser beam. When the laser pulse enters the water from the air, the volatility of the sea surface causes a change in the local normal vector, which affects the propagation direction of the laser pulse in the water and causes a displacement deviation of the submarine laser footprints. Here, the incident normal vector is $\vec{v}$, and the normal vector corresponding to the calm surface is $\vec{n} = (0,0,1)$. To facilitate the calculation, we set $|\vec{v}| = |\vec{n}| = 1$.

$$\cos \alpha = \frac{-\vec{n} \cdot \vec{L}}{|\vec{n}| \cdot |\vec{L}|} \tag{14}$$

$$\cos \beta = \sqrt{1 - \sin^2 \beta} = \sqrt{1 - \frac{1 - \cos^2 \alpha}{n_w^2}} \tag{15}$$

Figure 7 shows a schematic diagram of the laser refraction process. If we set $\vec{F_2} = \vec{f_1} + \vec{f_2}$ and $\vec{L} = \vec{l_1} + \vec{l_2}$, the relationship between the two vectors can be expressed as follows:

$$\begin{cases} \vec{l_1} = \vec{L} + \vec{n} \cdot \frac{|\vec{L}| \cos \alpha}{|\vec{n}|} \\ \vec{l_2} = -\vec{n} \cdot |\vec{L}| \cdot \frac{\cos \alpha}{|\vec{n}|} \\ \vec{f_1} = \frac{|\vec{F_2}| \cdot \vec{L}}{|\vec{L}| \cdot n_w} \\ \vec{f_2} = -\vec{n} \cdot |\vec{F_2}| \cdot \sqrt{1 - \frac{1 - \cos^2 \alpha}{n_w^2}} \end{cases} \tag{16}$$

The direction vector of laser propagation under the water surface is

$$\vec{F_2} = \left( \frac{\vec{L}}{|\vec{L}| \cdot n_w} - \vec{n} \cdot \sqrt{1 - \frac{1 - \cos^2 \alpha}{n_w^2}} \right) \cdot |\vec{F_2}| \tag{17}$$

Therefore, the key to accurately obtaining the incident direction of the target beam is to determine the inclination of the local water surface at the incident position in three-dimensional space and calculate the normal vector of the water surface undulations $\vec{v}$. We can replace $\vec{n}$ in Equation (17) with $\vec{v}$ and then correct the position of the underwater laser footprints based on the corresponding surface incidence position.

## 3. Experiments and Results

### 3.1. Simulated Surface Experiment

Ideally, the correction value of the incident angle should equal the sea surface inclination at the time of scanning, and the statistical characteristics of the corrected angles in the experimental areas should be consistent with the state of the surface undulations.

To quantitatively analyze the effects of the self-adaptive wave correction method, we first use a simulation experiment to process and compare the laser point clouds. The elevation of the water surface undulations is regarded as a function of the plane position $(x, y)$. A Gaussian mixture surface was used to simulate the wave surface in this section.

$$
\begin{aligned}
f(x,y) = \ & A\left[1 - \left(\tfrac{x}{m}\right)^2\right]\exp\left[-\left(\tfrac{x}{m}\right)^2 - \left(\tfrac{y}{n}+1.1\right)^2\right] \\
& - B\left[0.2 \times \tfrac{x}{m} - \left(\tfrac{x}{m}\right)^3 - \left(\tfrac{y}{n}\right)^5\right]\exp\left[-\left(\tfrac{x}{m}\right)^2 - \left(\tfrac{y}{n}\right)^2\right] \\
& + C\exp\left[-\left(\tfrac{x}{m}+1\right)^2 - \left(\tfrac{y}{n}\right)^2\right] \\
& - D\exp\left[-\left(\tfrac{x}{m}+1.2\right)^2 - \left(\tfrac{y}{n}\right)^2\right]
\end{aligned}
\tag{18}
$$

where $\{A, B, C, D\}$ are amplitude parameters of the wave surface model, and $\{m, n\}$ are range parameters. To accurately analyze the differences in normal vector estimation results under different conditions of surface elevation complexity, we use the wave surface simulation model above to simulate six surfaces with different elevation variation characteristics (S1~S6). S1~S6 in Figure 8 are simulated surfaces with different undulation characteristics that gradually change from gentle to violent.

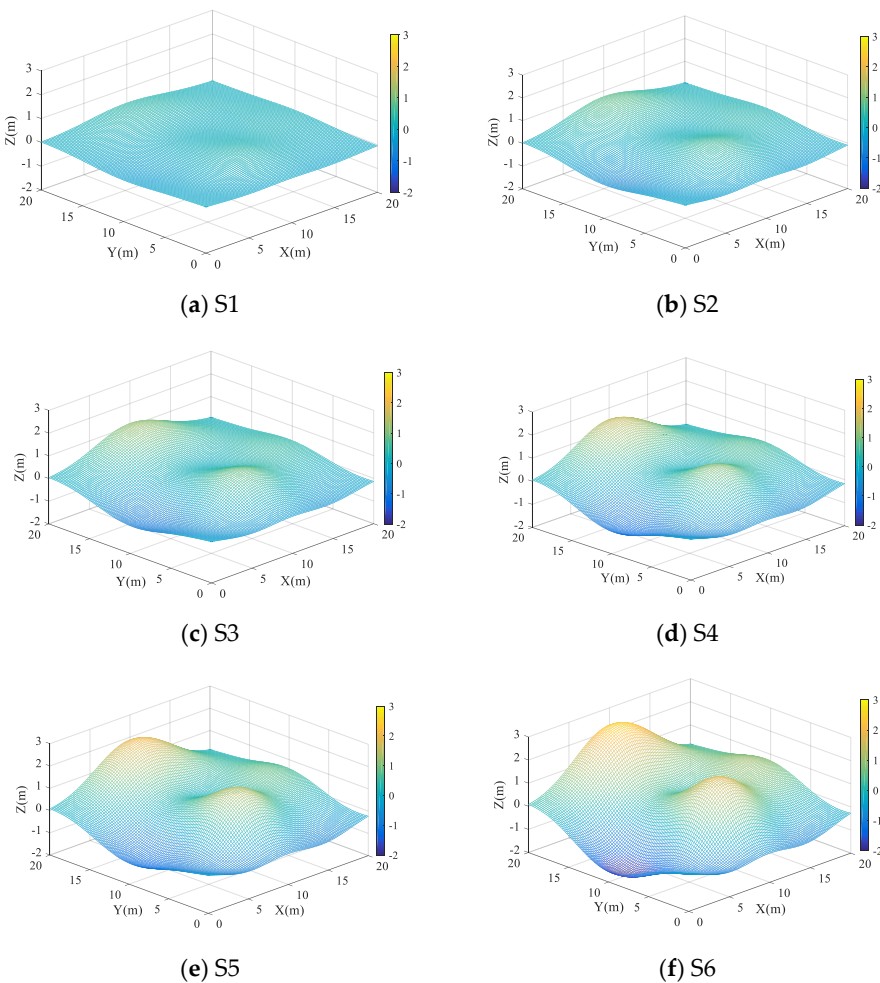

(**a**) S1         (**b**) S2

(**c**) S3         (**d**) S4

(**e**) S5         (**f**) S6

**Figure 8.** Simulated wave surfaces with different complexities.

Table 1 shows the specific parameters of the six simulated surfaces and their statistical features related to the undulation state. The *m* and *n* of each surface model are 30 and 28, respectively.

**Table 1.** Simulated surface parameters and complexity.

| Simulated Surface | Amplitude Parameter of Undulation | | | | Gradient of Elevation Change (cm) | | | | Slope (deg) | |
|---|---|---|---|---|---|---|---|---|---|---|
| | **A** | **B** | **C** | **D** | $\overline{f_x}$ | $\sigma_{fx}$ | $\overline{f_y}$ | $\sigma_{fy}$ | $\overline{S}$ | $\sigma_{slope}$ |
| S1 | 0.4 | 0.5 | 0.2 | 0.3 | 0.180 | 0.734 | 0.309 | 0.730 | 2.744 | 1.517 |
| S2 | 0.8 | 1.0 | 0.4 | 0.6 | 0.359 | 1.467 | 0.617 | 1.459 | 5.487 | 3.032 |
| S3 | 1.2 | 1.5 | 0.6 | 0.9 | 0.538 | 2.201 | 0.926 | 2.189 | 8.228 | 4.545 |
| S4 | 1.5 | 2 | 0.8 | 1.2 | 0.718 | 2.843 | 1.251 | 2.883 | 10.775 | 5.929 |
| S5 | 2.0 | 2.5 | 1.0 | 1.5 | 0.897 | 3.668 | 1.543 | 3.648 | 13.699 | 7.558 |
| S6 | 2.4 | 3.0 | 1.2 | 1.8 | 1.077 | 4.402 | 1.851 | 4.378 | 16.428 | 9.056 |

$\left\{\overline{f_x}, \sigma_{fx}\right\}$ and $\left\{\overline{f_y}, \sigma_{fy}\right\}$ are the mean value and root mean square error of the elevation gradient in the east and north directions, respectively, and $\left\{\overline{S}, \sigma_{slope}\right\}$ is the mean value and root mean square error of the undulating surface slope. The mean value and dispersion of the surface slope and elevation reflect the intensity of surface undulations. If the corresponding value is larger, the surface undulation is more intense. Table 2 shows the initial parameters of the wave correction method suggested in this paper.

**Table 2.** Initial parameters for dimension-based selection.

| Parameters | $r_0$ | $r_\varepsilon$ | $r_{max}$ |
|---|---|---|---|
| Value (m) | 1 | 0.25 | 3 |

The deviation of the laser propagation direction after refraction by the water surface waves mainly comes from the dip in the water surface normal vector in the vertical direction and its horizontal projection azimuth angle, which can be described by the slope and aspect of the local surface. Therefore, the accuracy of normal vector estimation can be evaluated by calculating and comparing the slope and aspect at the laser incident position.

The actual slope angle *S* and aspect angle *A* at position $(x, y)$ on the water surface can be calculated by the following equations:

$$S = \arctan\left(\sqrt{\left(\frac{\partial f}{\partial x}\right)^2 + \left(\frac{\partial f}{\partial y}\right)^2}\right) \tag{19}$$

$$A = 270° + \arctan\left(\frac{\partial f}{\partial x} / \frac{\partial f}{\partial y}\right) - 90°\left(\frac{\partial f}{\partial x} / \left|\frac{\partial f}{\partial x}\right|\right) \tag{20}$$

where $\frac{\partial f}{\partial x}$ and $\frac{\partial f}{\partial y}$ are the elevation gradients in the east and north directions, respectively. As the elevation changes in the simulated mathematical surface at the specific spatial position $(x, y)$ have a clear function, the accurate slope and aspect can be directly calculated by Equations (19) and (20).

According to the relevant values in Table 1 above, although the average value of the elevation gradient of each simulated surface in the east and north directions is different because S1~S6 are simulated mainly by the Gaussian function, the root mean square errors are relatively close in numerical value, and the complexity of the surface elevation gradually increases with the increase in surface amplitude parameters.

### 3.2. Natural Water Surface Experiment

According to Equation (17), the accuracy of the corrected positions of underwater laser points is closely related to the correction result of the normal vector at the laser incident position. When the accuracy of the normal vector is high, the surface undulation state reflected by normal vectors is more consistent with the actual situation. Furthermore, the accuracy of underwater terrain correction is more accurate. Under natural conditions, water surface undulations are mainly affected by complex dynamic environments, such as winds, swells, currents and coastal topography, which usually show obvious randomness. Therefore, we studied the correction results of the measured water surface undulations in experimental areas and verified the measured data processing effect of the proposed method in this section.

### 3.2.1. Experimental Areas

To verify the general applicability of this method, we selected two coastal areas as experimental areas. There are obvious differences in the geographical environment and data acquisition process in the two experimental areas. Experimental Area A is located in northwestern Wuzhizhou, which is an island belonging to Hainan Province, China (Figure 9b). The water depth is less than 20 m in the experimental area. The area has a sandy seafloor, and the underwater terrain varies with the seasons, but there are few reefs on the seabed. A Secchi disk was used to test the water transparency, and the result was approximately 7 m.

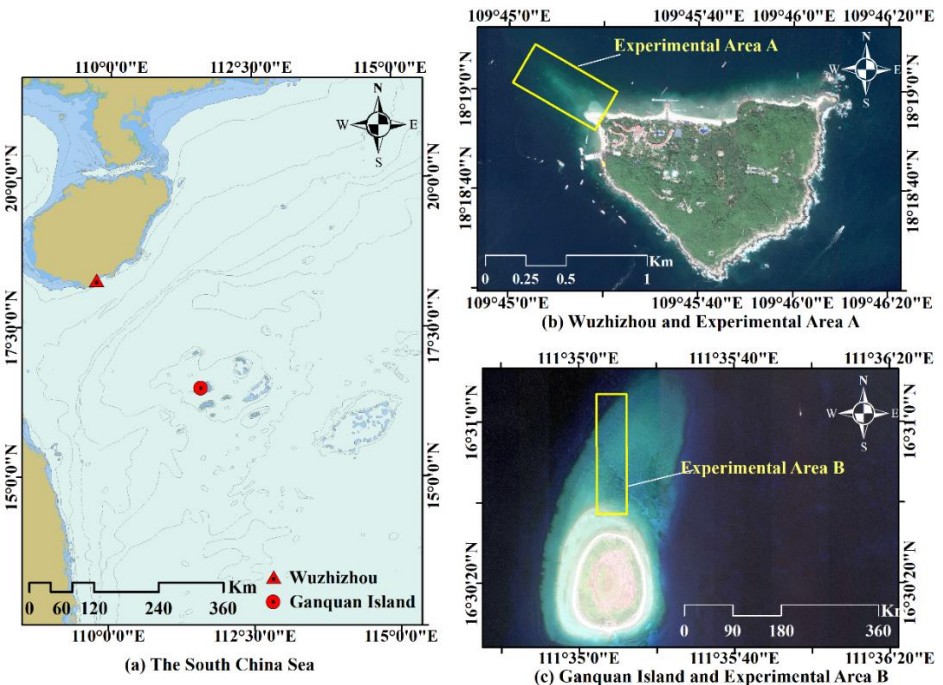

**Figure 9.** The experimental areas.

In January 2020, researchers from Shenzhen University used the single-wavelength ALB system iGreena (Figure 10a), which was researched and developed independently, to scan and detect Experimental Area A. A large amount of effective data was obtained. iGreena adopts an absolute circular scanning mode with a scanning field of view angle of 20°. The maximum detected depth is approximately 16 m.

Experimental Area B is located north of Ganquan Island, which is an island in the South China Sea (Figure 9c). Its sediment mainly consists of coral reefs, coral fragments and debris. At the end of December 2012, the First Institute of Oceanography (FIO), Ministry of Natural Resources of China, carried out a measurement experiment in the South China Sea

by using the Aquarius system (Figure 10b). Aquarius is a single-wavelength ALB system produced by Teledyne Optech and based on the Gemini system [42]. Aquarius adopts a linear scanning mode in which the laser points trace lines on the surface, and its scanning field of view angle is $\pm 20°$. The farthest offshore coverage by laser points reaches 1240 m in Experimental Area B. The Secchi disk transparency depth in the experimental area is approximately 8 m, and the maximum water depth obtained by the Aquarius system is approximately 16 m. The parameters in Table 3 are the main performance parameters for the two experimental systems.

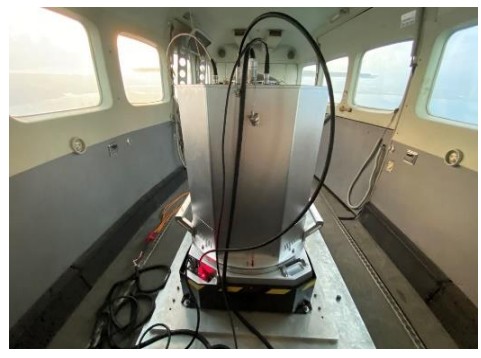
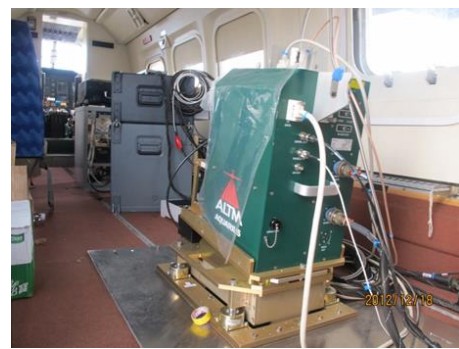

(**a**) iGreena          (**b**) Aquarius

**Figure 10.** ALB systems used in the experiment.

**Table 3.** Experimental system parameters.

| System Parameters | iGreena | Aquarius |
|---|---|---|
| Wavelength | 532 nm | 532 nm |
| Pulse repetition rate | 50~700 kHz | 33~70 kHz |
| Scan rate | 52 Hz | 70 Hz |
| Scan path | Circular | Linear |
| Scan half-angle | 20° | 0~±25° |
| Laser beam footprint | 131 mm@300 m, 219 mm@500 m | 300 mm@300 m, 500 mm@500 m |
| Acquisition frequency | 1.25 GHz | 1 GHz |
| Weight | 81 kg | 83 kg |

To verify the difference in the accuracy of the two ALB systems before and after wave correction, single-beam water depth detection instruments were used in both experiments to detect the water depth in relevant areas. The following are the parameters of the wave correction method used in the two experimental areas (Table 4).

**Table 4.** Initial parameters for dimension-based selection of neighbor points in the experiment.

| Experimental Area | $r_0$ | $r_\varepsilon$ | $r_{max}$ |
|---|---|---|---|
| Wuzhizhou | 1 | 1.5 | 10 |
| Ganquan Island | 1 | 1.5 | 10 |

As the Aquarius system adopts a linear scanning mode, the distribution of laser footprints in the detection area is more uniform than that from the iGreena system, which adopts a circular scanning mode. However, the scanning structure with high-frequency changes in the emission angle may have some disadvantages, such as a complex structure, the obvious influence of the direction of entry on echo signal intensity changes and a complex point cloud calculation model.

### 3.2.2. Consistency between Normal Vector Estimation Results and Natural Water Surface Undulations

Although relatively calm sea and good weather conditions were selected during the measurement operation, some obvious undulations can still be found on the surface point cloud. Figure 11 shows the surface undulation of the ocean point cloud obtained by ALB systems in the two experimental areas. In this section, we use several neighborhood selection radii to calculate the normal vector of the incident position in the two experimental areas, estimate the angle between the incident normal vector and the vertical direction, that is, the local sea surface slope angle of the water surface, and display them in respective statistical histograms.

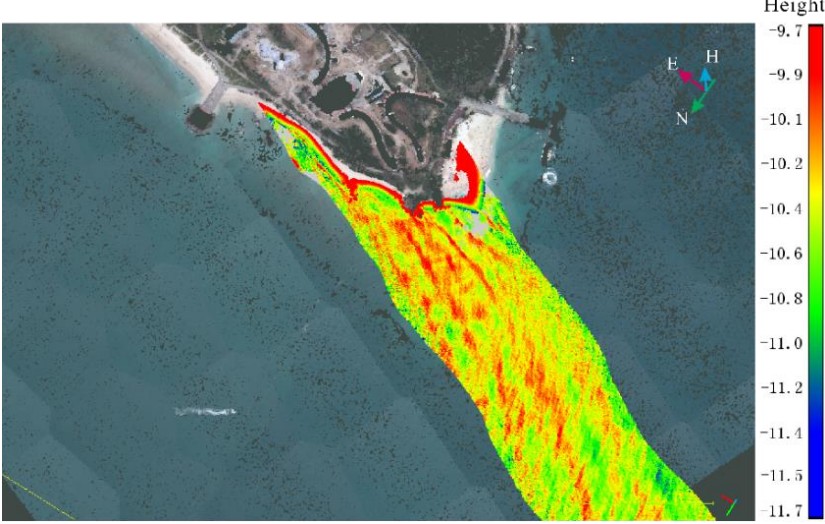

(**a**) The surface point cloud in Experimental Area A

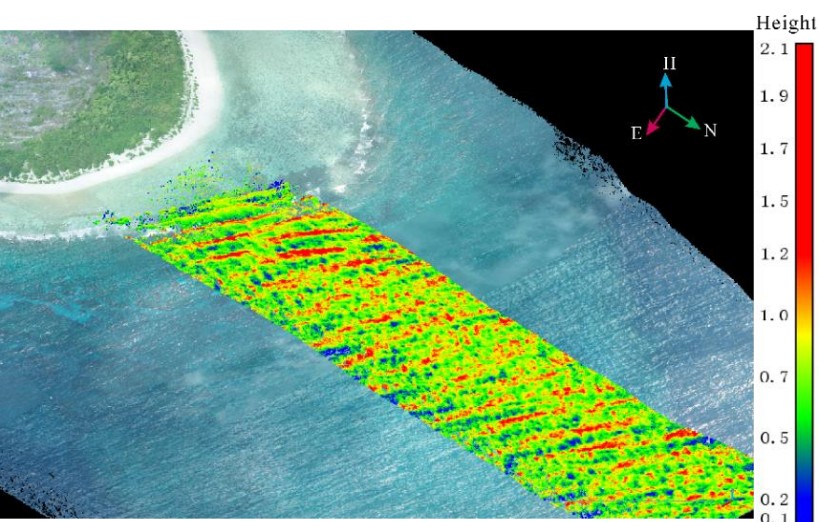

(**b**) The surface point cloud in Experimental Area B

**Figure 11.** The surface point cloud reflects the ocean state of the two experimental areas.

To facilitate the visualization of the distribution, a Weibull function was used to fit each probability distribution of the incident angle correction value. It can be clearly seen that the correction effect of the laser incident angle is different under different normal vector estimation ranges.

In Figures 12 and 13, the incidence angle correction value in Experimental Area A is mainly concentrated in the range of 0~30°, while the incidence angle correction value

in Experimental Area B is mainly concentrated in the range of 0–20°. In Figure 12, when the neighborhood radius is smaller, the incidence angle correction value is more scattered than the neighborhood radius is large, where the larger correction values come from high-frequency undulations and noise interference in the water surface point cloud. On the one hand, a small radius could reflect more details of the water surface wave; on the other hand, a radius that is too small could be easily affected by noise in the water surface point cloud, which could cause errors in the calculation of the incidence angle correction value. With the expansion of the normal vector estimation neighborhood, the waves whose half-lengths are less than the interval of laser points in the neighbor field are ignored, and the corresponding surface undulation is smoother than the actual water surface. Therefore, only the information of the waves with larger wavelengths could be retained in the processed data, and the probability density of the correction value is concentrated at smaller angles. In Figures 12a and 13a, the correction effects of the self-adaptive method of the dimensionality-based neighbor points are relatively moderate. Some microenvironment characteristics on the wave surface were retained, and the incidence angle estimation error caused by a neighborhood radius that was too small was avoided in some instances. In addition, the method suggested in this paper does not require point cloud interpolation in the normal vector estimation step, which is helpful to further simplify the data processing.

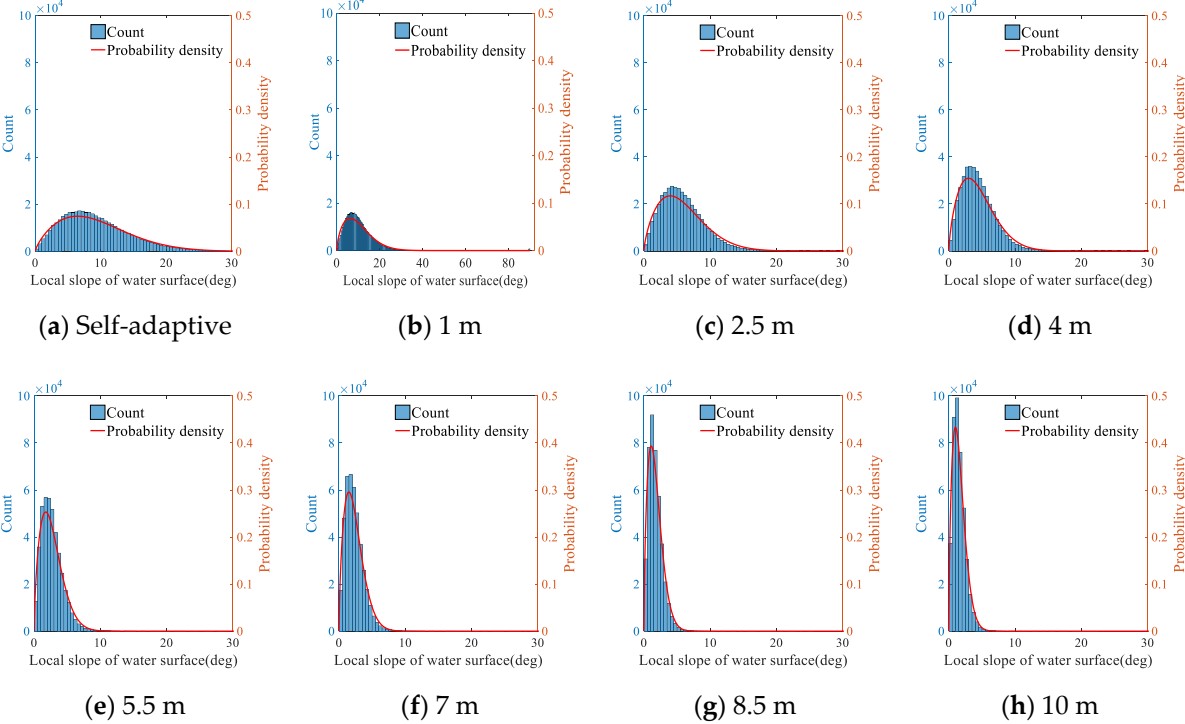

**Figure 12.** Statistical distribution of the incidence angle correction value using different neighborhood radii in Experimental Area A.

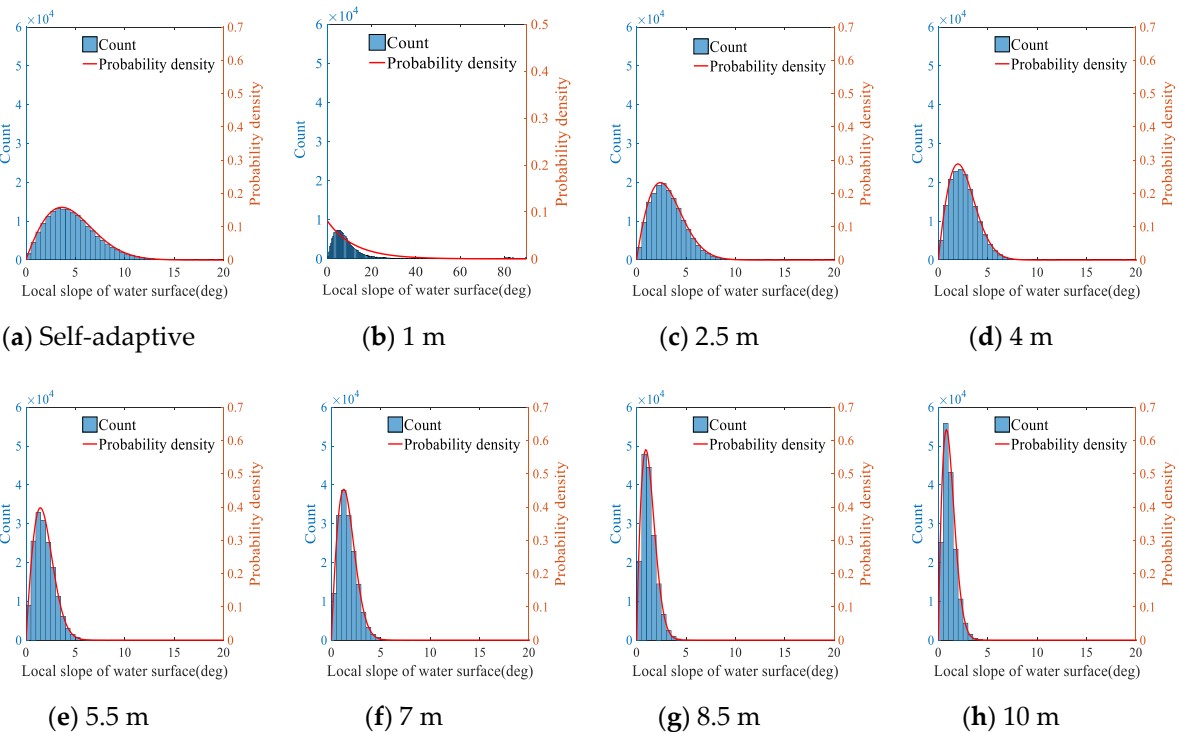

**Figure 13.** Statistical distribution of the incidence angle correction value using different neighborhood radii in Experimental Area B.

## 4. Discussion and Analysis

### 4.1. Accuracy Comparison for Surface Normal Vector Estimation

The ocean surface affected by wind waves and swells is usually complex and uncertain. The neighborhood point selection radius is directly related to whether the laser incidence correction method can reflect the features of the surface topography. The spatial direction of the laser incident normal vector can be decomposed into zenith and horizontal vectors (Figure 4). Ideally, their values should be equal to the surface slope and aspect at the laser incident position. In this section, we obtain the statistics of the slope and aspect of each simulated surface by normal vector estimation and compare them with the calculated values of the corresponding position of the surface model to obtain the different accuracies of the normal vector estimation results obtained by different methods.

In actual processing, the observation results of the elevation gradient in the east and north directions can be considered to have the same accuracy when laser detection equipment with stable performance is used, and the more violent the surface undulation in a fixed area is, the greater the root mean square error (RMSE) $\sigma_f$.

$$\sigma_S = \cos^2 S \cdot \sigma_f \tag{21}$$

$$\sigma_A = \cot S \cdot \sigma_f \tag{22}$$

Equations (21) and (22) indicate that the main factors affecting the accuracy of the surface slope and aspect estimation are the detection accuracy of the local surface slope and elevation change at the spatial position.

In the following sections, the difference between the dimension-based self-adaptive neighborhood point selection method and the fixed neighborhood point selection method is compared intuitively and quantitatively. The relationship between the neighborhood size and the accuracy of normal vector estimation under different surface changes will be analyzed in detail.

(1)  The effects of surface inclination and neighborhood point selection radius on normal vector estimation accuracy

To analyze the influence of the neighborhood point selection radius and the surface inclination on the accuracy of normal vector estimation, we selected the simulation surface S4 (Figure 8d) as the research object, and the classification statistics of the results by different normal vector estimation radii and surface slope angles were calculated. Figures 14 and 15 show the statistical results of the estimated error of the surface slopes and aspects, and Tables 5 and 6 show the specific values.

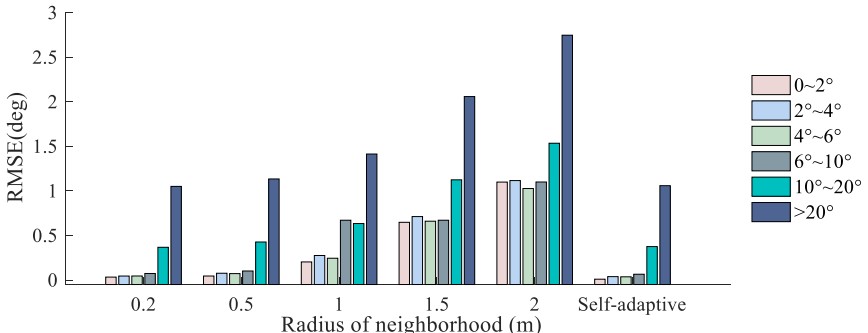

**Figure 14.** Accuracy of surface slope estimation with grades of slope value in different neighborhood point selection radii.

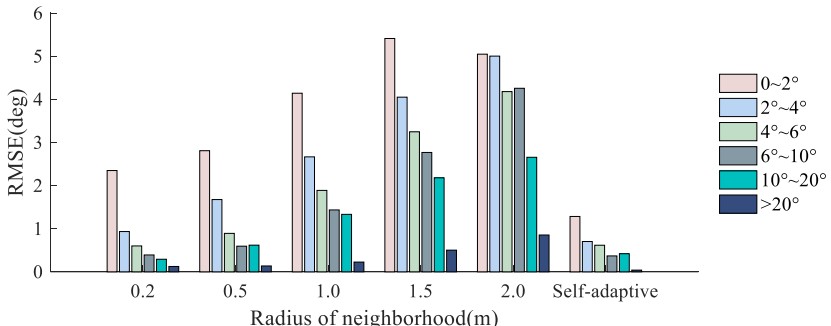

**Figure 15.** Accuracy of surface aspect estimation with different grades of slope and neighborhood point selection radii.

**Table 5.** Root mean square error of wave surface slope estimation in different intervals.

| Radius of Neighborhood (m) | Surface Slope Interval | | | | | | |
|---|---|---|---|---|---|---|---|
| | 0~2° | 2°~4° | 4°~6° | 6°~10° | 10°~20° | >20° | >0° |
| 0.2 | 0.033 | 0.045 | 0.046 | 0.074 | 0.368 | 1.051 | 0.279 |
| 0.5 | 0.045 | 0.076 | 0.072 | 0.102 | 0.427 | 1.135 | 0.325 |
| 1 | 0.203 | 0.276 | 0.245 | 0.671 | 0.634 | 1.414 | 0.633 |
| 1.5 | 0.648 | 0.712 | 0.660 | 0.671 | 1.125 | 2.059 | 0.987 |
| 2 | 1.099 | 1.116 | 1.027 | 1.100 | 1.535 | 2.747 | 2.505 |
| Self-adaptive | 0.010 | 0.039 | 0.037 | 0.065 | 0.376 | 1.059 | 0.274 |

From the overall trend of the error statistics in Figures 14 and 15, the radius of neighborhood point selection has an obvious correlation with the correction accuracy. In the case of similar surface inclinations, the estimated deviation based on the point clouds is positively correlated with the radius of the neighborhood point selection, while the estimation accuracy of the laser incident deviation decreases with increasing neighborhood point radius. Actually, a larger inclination usually occurs in areas with violent elevation

changes, and the estimation accuracy of the local laser incident surface is affected by the coupled effects of the surface elevation and inclination. Judging from the numerical values of the experimental results, when the local water surface inclinations are similar, the deviation in the aspect estimation is larger than the deviation in the slope estimation. Within the same normal vector estimation neighborhood radius, the slope estimation accuracy gradually decreases with increasing surface inclination, which means that the main factor that affects the slope estimation is the elevation gradient detection accuracy of the ALB system. Unlike the case of slope estimation, the key factor that affects the accuracy of local aspect estimation is the surface inclination.

**Table 6.** Root mean square error of wave surface aspect estimation in different intervals.

| Radius of Neighborhood (m) | Surface Slope Interval | | | | | | |
|---|---|---|---|---|---|---|---|
| | 0~2° | 2°~4° | 4°~6° | 6°~10° | 10°~20° | >20° | >0° |
| 0.2 | 2.349 | 0.934 | 0.600 | 0.391 | 0.292 | 0.124 | 0.439 |
| 0.5 | 2.809 | 1.676 | 0.892 | 0.595 | 0.619 | 0.136 | 0.721 |
| 1 | 4.142 | 2.666 | 1.888 | 1.435 | 1.333 | 0.226 | 1.461 |
| 1.5 | 5.412 | 4.051 | 3.248 | 2.768 | 2.180 | 0.503 | 2.463 |
| 2 | 5.048 | 5.003 | 4.180 | 4.257 | 2.657 | 0.854 | 3.259 |
| Self-adaptive | 1.284 | 0.703 | 0.617 | 0.369 | 0.420 | 0.040 | 0.435 |

(2)  The effects of elevation undulations and neighborhood point selection radii on normal vector estimation accuracy

The RMSEs for the slopes and aspects obtained by different neighborhood selection schemes are shown in Figures 15 and 16, and the specific values are in Tables 7 and 8.

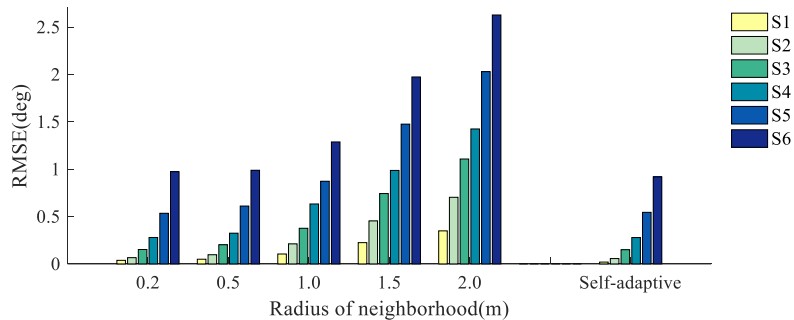

**Figure 16.** Accuracy of surface slope estimation with different surface complexities and selection ranges of neighborhood points.

**Table 7.** Root mean square error of wave surface slope estimation.

| Selection Radius of Neighborhood (m) | Simulated Wave Surfaces | | | | | |
|---|---|---|---|---|---|---|
| | S1 | S2 | S3 | S4 | S5 | S6 |
| 0.2 | 0.038 | 0.066 | 0.152 | 0.279 | 0.535 | 0.974 |
| 0.5 | 0.050 | 0.097 | 0.203 | 0.325 | 0.611 | 0.989 |
| 1 | 0.105 | 0.212 | 0.377 | 0.633 | 0.873 | 1.288 |
| 1.5 | 0.225 | 0.454 | 0.743 | 0.987 | 1.476 | 1.974 |
| 2 | 0.349 | 0.704 | 1.108 | 1.425 | 2.030 | 2.628 |
| Self-adaptive | 0.019 | 0.057 | 0.150 | 0.279 | 0.545 | 0.919 |

From Figures 16 and 17, it can be seen that the simulated surface elevation fluctuations have a significant impact on the estimation accuracy of the local slope and aspect. Moreover, the correlation between the estimation accuracy and the radius of the neighborhood is consistent with the previous discussion. With the same neighborhood radius, the slope

calculation accuracy in a flat point cloud surface is relatively high, while the calculation error of the local aspect gradually increases with the decrease in the elevation gradient.

**Table 8.** Root mean square error of wave surface aspect estimation.

| Selection Radius of Neighborhood (m) | Simulated Surface | | | | | |
|---|---|---|---|---|---|---|
| | S1 | S2 | S3 | S4 | S5 | S6 |
| 0.2 | 1.136 | 0.661 | 0.510 | 0.439 | 0.400 | 0.522 |
| 0.5 | 1.264 | 0.865 | 0.777 | 0.721 | 0.680 | 0.693 |
| 1.0 | 1.857 | 1.669 | 1.642 | 1.461 | 1.520 | 1.630 |
| 1.5 | 2.542 | 2.449 | 2.413 | 2.463 | 2.376 | 2.367 |
| 2.0 | 3.287 | 3.210 | 3.200 | 3.259 | 3.164 | 3.121 |
| Self-adaptive | 0.423 | 0.433 | 0.447 | 0.435 | 0.442 | 0.481 |

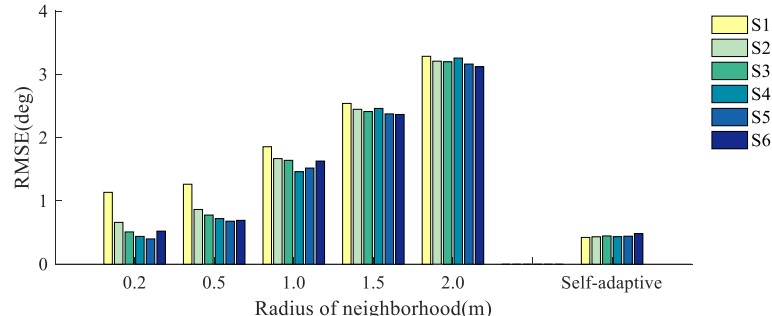

**Figure 17.** Accuracy of surface aspect estimation with different surface complexities and selection ranges of neighborhood points.

In the above simulated surface experiments, the dimension-based neighbor point selection method suggested in this paper maintains good slope and aspect estimation accuracy under different elevation fluctuations. The experimental results reflect the following two features in the laser incident normal vector correction process:

a. Adaptability of the incidence angle correction method

According to the previous analysis, we can infer that the main factor affecting the slope estimation accuracy is the surface elevation complexity, and the surface inclination at the target position plays a significant role in the accuracy of surface aspect estimation.

In Figures 14–17, the RMSE of the estimated slope and aspect deviation increases with increasing neighborhood selection radius under different changes in surface inclination and elevation. This is because when the neighborhood radius increases, more laser points participate in the calculation of normal vectors. If the laser point selection area exceeds the region of terrain features, then a smoothing effect is produced in the processing of normal vector estimation based on PCA.

The undulation characteristics reflected by the surface point cloud obtained by scanning have been limited by time and space. When the neighborhood radius used in the estimation of the incident normal vector is too large, it often results in distortion of the incidence correction effect. When the neighborhood is too small, the noise in the water surface point cloud can easily interfere with the correction result. Adopting a neighborhood selection strategy with an adaptive mechanism is beneficial to improve the applicability of the water surface incidence correction method to different systems and environmental conditions. Through the above experiments, it can be seen that the dimension-based neighborhood selection method can adjust the selection radius of the corresponding neighborhood based on the principle of information entropy, and the calculation process has good adaptability.

b. Stability of the correction effect under different sea conditions

In Equations (22) and (23), $\cos^2 S$ is a monotonically decreasing function, and its value is less than 1. Compared with that of $\cot S$ in Equation (23), which is also a monotonically decreasing function, the descent speed of $\cos^2 S$ is significantly lower. Therefore, the accuracy of slope aspect estimation results is more sensitive to slope changes. By analyzing the phenomena in Figures 16 and 17 above, it can be seen that the influence of the surface elevation complexity on the accuracy of the surface slope estimation is greater than that of the slope value in a fixed area, but for the surface aspect, the degree of the surface inclination is the main factor affecting the accuracy. In addition, the wave undulations on the sea surface are complex and changeable. Different scanning methods and environmental conditions affect the density of the point cloud of the water surface. This should be the direct reason for the different effects of laser incidence correction methods. Since the method suggested in this paper can adaptively select the neighborhood radius through surface morphology characteristics, it maintains good normal vector estimation accuracy for different surface undulations, which is of great significance for the incidence correction stability under different surface fluctuations.

*4.2. Correction Effect for ALB System Measured in the Areas*

The uneven distribution of the water surface point cloud caused by the scanning mode of airborne LiDAR bathymetry and the dynamic water environment in the target area has a great impact on the estimation of the normal vector of the incident laser. To analyze the influence of the neighborhood point selection radius at the incident position for estimating the normal vector, different neighborhood radii were used to calculate the normal vector based on the actual measured surface point clouds in Experimental Area A and Experimental Area B.

a.      Estimation of the slope of the wind wave surface

To evaluate the difference in the effect of the laser incident normal vector estimation with different neighborhood point selection radii on the actual wave surface, we calculated the correction angles of the laser incident normal vectors, which can be regarded as the surface slopes, in the different experimental areas and then used the Weibull function to fit the values of the correction angles in Figures 12 and 13.

$$f(x; \lambda, k) = \begin{cases} \frac{k}{\lambda}\left(\frac{x}{\lambda}\right)^{k-1} \exp\left[-\left(\frac{x}{\lambda}\right)^k\right] & x \geq 0 \\ 0 & x < 0 \end{cases} \tag{23}$$

where $x$ is the random variable, $\lambda$ is the scale parameter and $k$ is the shape parameter. In the Weibull function, if the shape parameter is increasing, the peak value of the probability density function deviates from zero, while the larger the scale parameter is, the wider the probability density function. The trends of the scale and shape parameters of the results obtained from the processing of each neighborhood radius are shown in Figure 18 and Table 9.

In Figure 18 and Table 9, there is no obvious correlation between the values of the scale parameter and the neighborhood radius, and the change in the scale parameter magnitude is small. This indicates that the dispersion of the water surface slope does not change much for different radii. In addition, the shape parameter decreases with increasing radius, which indicates that the maximum probability of sea surface inclination gradually approaches zero; that is, as the local normal vector estimated neighborhood radius increases, the laser incidence angle compensation value decreases. This is because a larger radius causes local undulation characteristics to be ignored in the process of the incidence normal vector calculation, so the water surface state described by the normal vectors tends to be smooth. Since laser incidence normal vector estimation using a smaller neighborhood radius is generally susceptible to abrupt local changes and surface signal noise interference, the undulating characteristics of the water surface cannot be accurately reflected in general. This result is consistent with the correspondence between the estimated water surface slope and the maximum probability density.

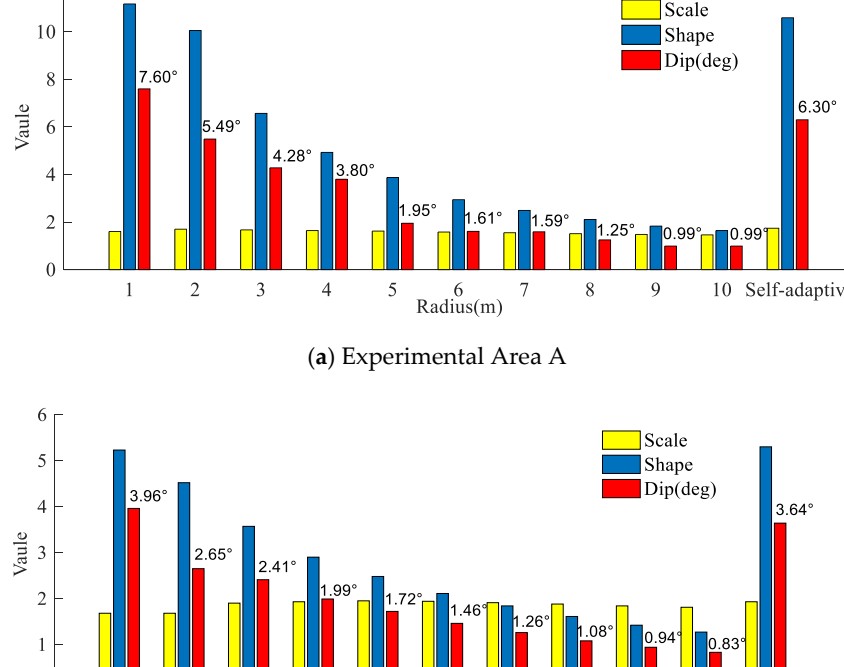

(**a**) Experimental Area A

(**b**) Experimental Area B

**Figure 18.** Parameters of the Weibull function fitting for the water surface inclination estimation with different neighborhood point selection radii in the experimental areas.

**Table 9.** Fitting parameters for the Weibull functions with different neighborhood point selection radii.

| Radius (m) | Wuzhizhou | | | Ganquan Island | | |
|---|---|---|---|---|---|---|
| | Scale | Shape | $S_{w\_max}$ (deg) | Scale | Shape | $S_{w\_max}$ (deg) |
| 1 | 1.60 | 11.17 | 7.60 | 1.68 | 5.23 | 3.96 |
| 2 | 1.70 | 10.05 | 5.49 | 1.68 | 4.52 | 2.65 |
| 3 | 1.67 | 6.57 | 4.28 | 1.90 | 3.57 | 2.41 |
| 4 | 1.64 | 4.93 | 3.80 | 1.93 | 2.90 | 1.99 |
| 5 | 1.62 | 3.87 | 1.95 | 1.95 | 2.48 | 1.72 |
| 6 | 1.58 | 2.94 | 1.61 | 1.94 | 2.11 | 1.46 |
| 7 | 1.55 | 2.49 | 1.59 | 1.91 | 1.84 | 1.26 |
| 8 | 1.51 | 2.11 | 1.25 | 1.88 | 1.61 | 1.08 |
| 9 | 1.48 | 1.83 | 0.99 | 1.84 | 1.42 | 0.94 |
| 10 | 1.46 | 1.64 | 0.99 | 1.81 | 1.27 | 0.83 |
| Self-adaptive | 1.74 | 10.59 | 6.30 | 1.93 | 5.30 | 3.64 |

Local ocean surface wind waves always have a certain randomness. Let the surface inclination be $S$, and $v$ is the surface wind speed. The probability density $p$ of the ocean surface inclination can be calculated according to the Cox–Munk model. The model is as follows:

$$p(S) = \frac{2}{\sigma^2} \exp\left(-\frac{\tan^2 S}{\sigma^2}\right) \tan S \sec^2 S \tag{24}$$

$$\sigma^2 = 0.003 + 0.00512v \tag{25}$$

where the inclination angle corresponding to the maximum probability density is $S_{p\_\text{max}}$, and its value reflects the overall features of the sea surface undulation state under the current wind speed conditions.

We analyze the consistency of the local undulating water surface inclination and the incidence correction results in the following. The surface inclination angles can be estimated from wind speed reanalysis data, and the Cox–Munk model and the local laser incident angles can be calculated from the surface laser point clouds obtained by different ALB systems in the two experimental areas. Since Experimental Area A is only 3 km away from the mainland of Hainan Island, considering that the low-altitude wind speed in this area is obviously affected by the land of Hainan Island, we choose the wind speed reanalysis data of ERA5-Land of the European Centre for Medium-Range Weather Forecasts (ECMWF) as a reference (https://cds.climate.copernicus.eu/cdsapp#!/dataset/reanalysis-era5-land?tab=form, accessed data: 13 August 2020). The time resolution of the ERA5-Land reanalysis data is 1 hour, the spatial resolution is 0.1° in latitude and longitude and its numerical results take into account the interaction of wind speeds on the underlying surface, which has good applicability in land and coastal areas. Experimental Area B lacks reanalysis data with underlying surface analysis. Therefore, we use National Centers for Environmental Prediction (NCEP) wind speed data as a reference for comparison (ftp://ftp.cdc.noaa.gov/Datasets/ncep.reanalysis/surface_gauss, accessed data: 13 August 2020). The time resolution of NCEP reanalysis data is 1 hour, and the spatial resolution is 0.25° in latitude and longitude.

Table 10 shows the sea surface wind speed data of the two experimental areas, where u10 is the northward component of the 10 m wind, v10 is the eastward component of the 10 m wind and GMT is Greenwich Mean Time.

**Table 10.** The 10-m wind speeds and the probability extremum $S_{p\_\text{max}}$ in two experimental areas.

| Experimental Area | Location | GMT | Speed (ms$^{-1}$) | | | $S_{p\_\text{max}}$ (deg) |
|---|---|---|---|---|---|---|
| | | | u10 | v10 | $\sqrt{(u10)^2+(v10)^2}$ | |
| A | E109.70° N18.3° | 2020-01-10 9:00 2020-01-10 10:00 | −4.35 −4.10 | −0.14 −0.33 | 4.36 4.11 | 6.50 6.33 |
| B | E111.25° N16.25° | 2012-12-22 6:00 2012-12-22 7:00 | −6.97 −7.30 | −7.09 −6.86 | 9.94 10.02 | 7.54 7.57 |

During the experimental period, the wind speed at 10 m above the water surface was approximately 4.11 m/s, and the wind speed above Ganquan Island was approximately 9.94 m/s. However, because Experimental Area B is in the semi-enclosed area formed by coral reefs in the Xisha Yongle Atoll area, the water surface has a small degree of wave undulation. The deviation between the sea surface inclination corresponding to the largest probability density calculated by the Cox–Munk model and the correction value of the laser incident angle obtained by using different neighborhood radii can be calculated as follows:

$$\Delta s = |S_{w\_\text{max}} - S_{p\_\text{max}}| \tag{26}$$

Based on the above equation, the variations in the deviation between $S_{p\_\text{max}}$ obtained from the wind speed data during the experimental period and $S_{w\_\text{max}}$ in Table 10 are shown in Figure 19.

Figure 19 shows the surface undulations under different neighborhood selection conditions. Compared with Experimental Area A, Experimental Area B is a part of the Yongle Atolls in the South China Sea, and the actual wind and waves on the surface of the atoll are limited by the water depth and seafloor topography. Therefore, although the deviation of Experimental Area B is larger than that of Experimental Area A, the trends of the surface undulation estimation results of the two experimental areas are similar. In Figures 18 and 19, as the neighborhood is too small, the stability of the laser incident normal

vector estimation effect is poor. This phenomenon may be caused by the interference of water surface undulations and local point cloud noise. When the neighborhood is too large, the slope estimation result from surface point clouds is more distinct from that of the Cox–Munk model. The self-adaptive neighborhood selection method based on the spatial distribution of point clouds and the principle of information entropy effectively avoids the difference caused by different neighborhood selection ranges in the normal vector calculation process. The result processed by this self-adaptive method shows that it can maintain local undulation details and restrain the abrupt change in water surface elevation to some extent.

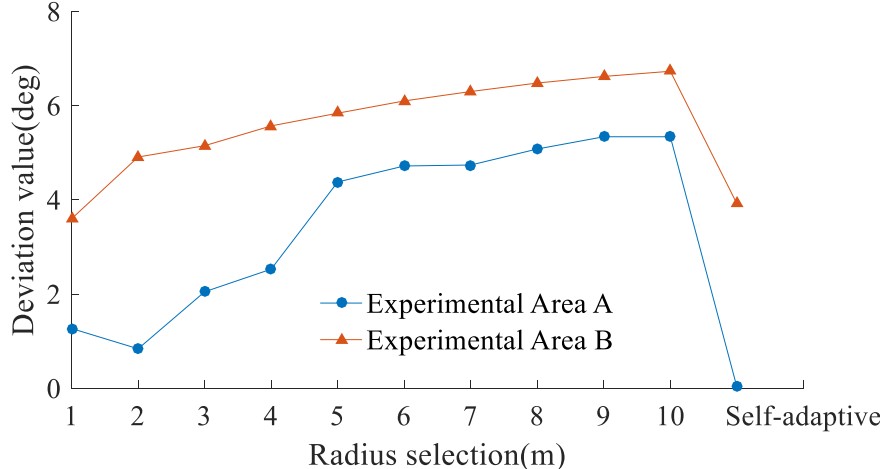

**Figure 19.** The deviation between the values of the incidence angle correction and the surface probability extremum slopes in the two experimental areas.

b.    The improvement of laser incident correction on bathymetry accuracy

To evaluate the ALB detection accuracy, we simultaneously carried out single-beam water depth detection experiments. The spatial distribution of single-beam depth points in the experimental areas is shown in Figure 20.

The vertical deviation between the synchronously observed single-beam depth points and the mesh established by the laser point cloud on the seafloor can be used to evaluate the water depth detection effect after the undulation surface incidence correction. Based on this, we analyzed the improvement of the laser detection results at specific locations.

In Figure 20, the distribution of the depth points evenly covers the ranges of the experimental areas, although the sampling frequency of the single-beam depth points in Experimental Area A is relatively low. The histograms of the deviation between the mesh of the seafloor laser point cloud and the single-beam depth points are shown in Figure 21.

Figure 21 shows that the discrete state of the deviation values in the initial data has been improved, and most deviation values are concentrated near the value of zero, which means that the deviation between the point cloud and the single beam after the sea surface wave correction is significantly reduced.

The single-beam bathymetrical points in Experimental Area B evenly cover the experimental area according to the designed route, and more depth points are obtained by improving the sampling frequency.

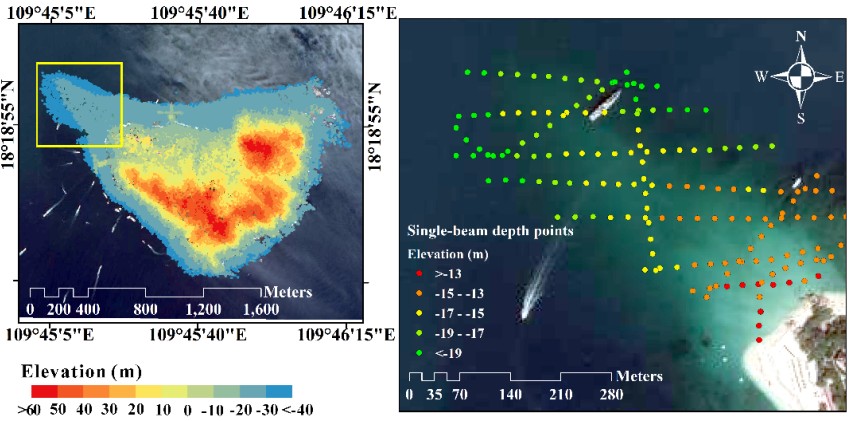

(**a**) Single-beam depth points in Experimental Area A

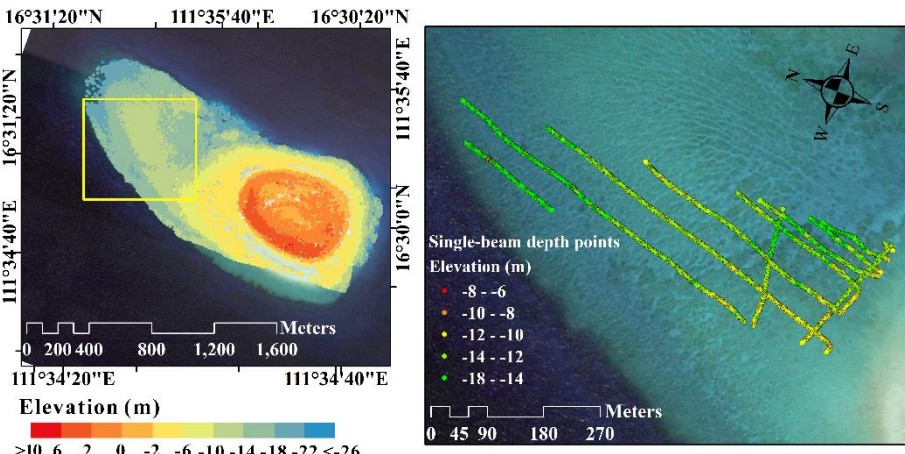

(**b**) Single-beam depth points in Experimental Area B

**Figure 20.** The distribution of the single-beam depth points in the experimental areas.

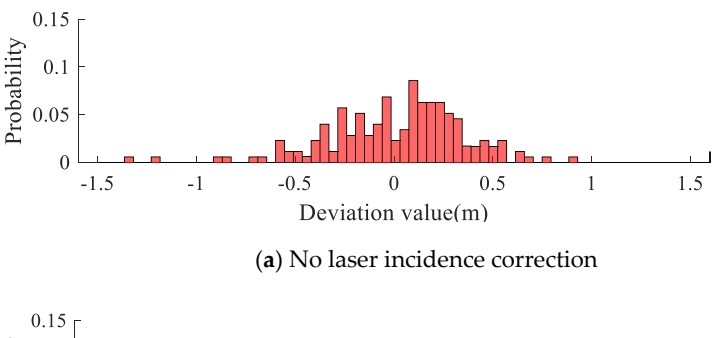

(**a**) No laser incidence correction

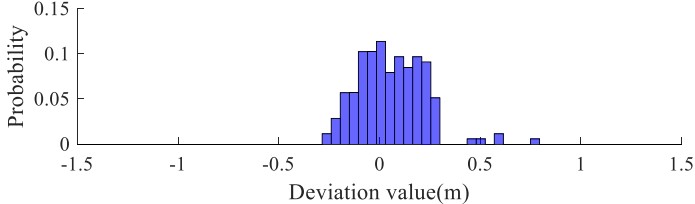

(**b**) Laser incidence correction using the method in this paper

**Figure 21.** Probability distribution of depth deviation before water surface deviation correction vs. single-beam depth data in Experimental Area A.

The statistical result of Experimental Area B in Figure 22 also shows a similar characteristic to that of Experimental Area A after laser surface incident correction. Table 3 contains the average and the RMSE of deviations between the correction result and the single-beam depth point in the two experimental areas. In Equations (27) and (28), $D_{Sb}$ is the depth value of a single beam on the seafloor, and $D_{ALB}$ is the depth value of the seafloor mesh obtained by ALB at the corresponding position. $N$ is the number of the single-beam depth points. $m_{average}$ is the average of the deviation value, and $m_{rmse}$ is the RMSE.

$$m_{average} = \frac{1}{N-1} \sum_{i=1}^{N} (D_{ALB} - D_{Sb}) \tag{27}$$

$$m_{rmse} = \sqrt{\frac{1}{N-1} \sum_{i=1}^{N} (D_{ALB} - D_{Sb})^2} \tag{28}$$

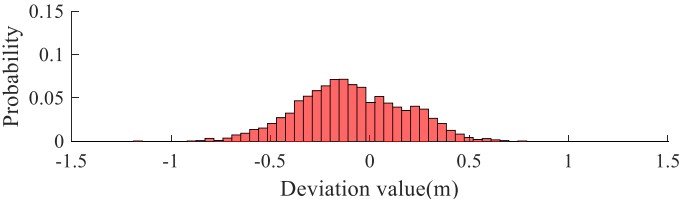

(**a**) No laser incidence correction

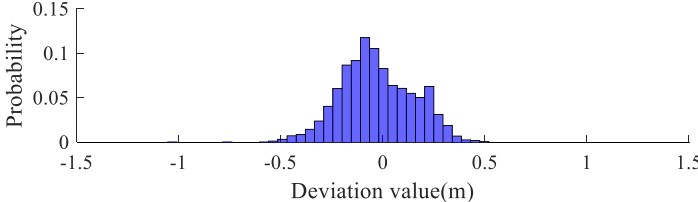

(**b**) Laser incidence correction using the method in this paper

**Figure 22.** Probability distribution of depth deviation before water surface deviation correction vs. single-beam depth data in Experimental Area B.

From the statistics of the deviation in Table 11, the sea surface wave correction based on the dimension-based self-adaptive local normal vector calculation can shift the results obtained by the ALB system closer to the true water depths, where the reduction is the absolute value of the change before and after the incident correction. After laser incidence correction, the depth detection results in the experimental area met the accuracy requirements of International Hydrographic Organization Special Order (IHO S-44) [43]. According to the experimental analysis, this model is an effective sea surface wave correction model that can effectively solve the uncertainty caused by sea surface undulations in the airborne laser detection process.

**Table 11.** Effect comparison before and after laser incident correction.

| Statistics | Experimental Area A (iGreena) | | | Experimental Area B (Aquarius) | | |
|---|---|---|---|---|---|---|
| | **Before** | **After** | **Reduction** | **Before** | **After** | **Reduction** |
| $m_{average}$ (m) | 0.288 | 0.134 | 0.154 | −0.241 | −0.154 | 0.087 |
| $m_{rmse}$ (m) | 0.401 | 0.174 | 0.227 | 0.350 | 0.229 | 0.121 |

Previous studies on laser incident correction methods for small-footprint ALB systems mainly referred to local curved surface modeling to reconstruct the laser propagation

process. Complex modeling based on scanning point clouds for instantaneous fluctuating water surfaces is the focal point for such methods [44,45]. In [12], the water and land integration point cloud around Wuzhizhou collected by the Aquarius system has the same sediment as and similar environmental conditions to Experimental Area A in this article. Yang et al. used the wave spectrum model to fit the air–water interface and reduced the error from laser refraction, and the RMSE decreased by 9.2 cm. However, the wave surface simulation method aims to reconstruct the laser incident surface by building the wave spectrum model, which relies on the ALB system scan status and ocean environment-related parameter presets. Therefore, the correction effect is obvious, but the process is more complicated.

The method proposed in this paper can estimate the laser incident direction in a self-adaptive way, and the correction accuracy of the water depth detection results is good and relatively stable, especially under the conditions of complex ocean surface undulations and uneven points. In addition, self-adaptive processing based on the principles of information entropy and local interface feature dimensions does not require a complicated sea surface simulation, which is beneficial in fast processing.

### 4.3. The Influence of Undulating Water on ALB Depth Detection

In addition to the signal reception and data processing accuracy of the system itself, the factors directly affecting the water depth detection capability of the ALB system also include the zenith angle of the scanning laser beam, the refractive index of the propagation medium and the water depth conditions of the target area. The dimension-based self-adaptive method of laser incidence correction can reduce the negative influence of ocean surface waves on ALB system bathymetric detection. If we regard the influence of the undulating water surface on the incident angle as a kind of incident angle error, the depth deviation caused by the change in the incident direction can be analyzed according to the law of error propagation. $D$ in Equation (29) is the instantaneous depth obtained by the ALB system after signal processing.

$$D = \left( \frac{H}{|\vec{L}| \cdot n_w} - \frac{\cos\alpha - \sqrt{n_w^2 - 1 + \cos^2\alpha}}{n_w} \right) \cdot |\vec{F_2}| \tag{29}$$

The water depth obtained by the ALB system is mainly related to the altitude of the airborne platform $H$, the laser emission angle $\alpha$, the relative refractive index $n_w$ between the air and water and the propagation distance of the laser in the air and water, $|\vec{L}|$ and $|\vec{F_2}|$, respectively. $H$ is detected by the position and orientation system (POS), and the accuracy of $|\vec{L}|$ and $|\vec{F_2}|$ of each laser beam mainly depends on the echo signal processing capability. In addition, because the change in the optical refractive index is relatively small in shallow and uniform water, the influence on the laser underwater detection result itself is smaller than that from the other factors proposed above, and its influence will not be analyzed in much detail here.

If only the environmental impact of the detection area is considered, the change in the angle of incidence caused by the real-time changes in water surface waves should be regarded as an important environmental factor that causes the system's bathymetric deviation. In the above equation, according to the law of error propagation, the water depth detection accuracy model of the beam incidence deviation caused by surface undulation can be calculated.

$$\sigma_D = \sqrt{\left[ \sin\alpha - \frac{\cos\alpha \sin\alpha}{\sqrt{n_w^2 + \cos^2\alpha - 1}} \right] \frac{D}{\cos\beta} \cdot \sigma_\alpha} \tag{30}$$

where $\sigma_\alpha$ is the deviation of the laser incident angle caused by the undulation of the water surface, which is equal to the slope angle of the local surface, and $\beta$ is the refraction

angle of the laser beam after passing through the water surface. The specific value can be calculated according to Equation (30). This reflects the relationship between the system's underwater terrain detection accuracy and the laser incident angle. Based on this, we calculated the depth deviation of the ALB system with different incident angles and water depth conditions.

Figure 23 indicates that the deeper the target ocean area is, the more susceptible the ALB system is to surface undulations. When considering only the impact of surface waves, the bathymetric accuracy of the ALB system also has significant differences in the responses to different emission angles and ocean surface slopes. In recent years, commercial ALB systems have mainly used the circular scanning method with a single emission angle, and the laser emission angle is in the range of 15–20°. As the linear scanning method constantly changes the emission angles during scanning processing, the circular scanning method has more advantages in the overall accuracy and stability of the depth estimation deviation.

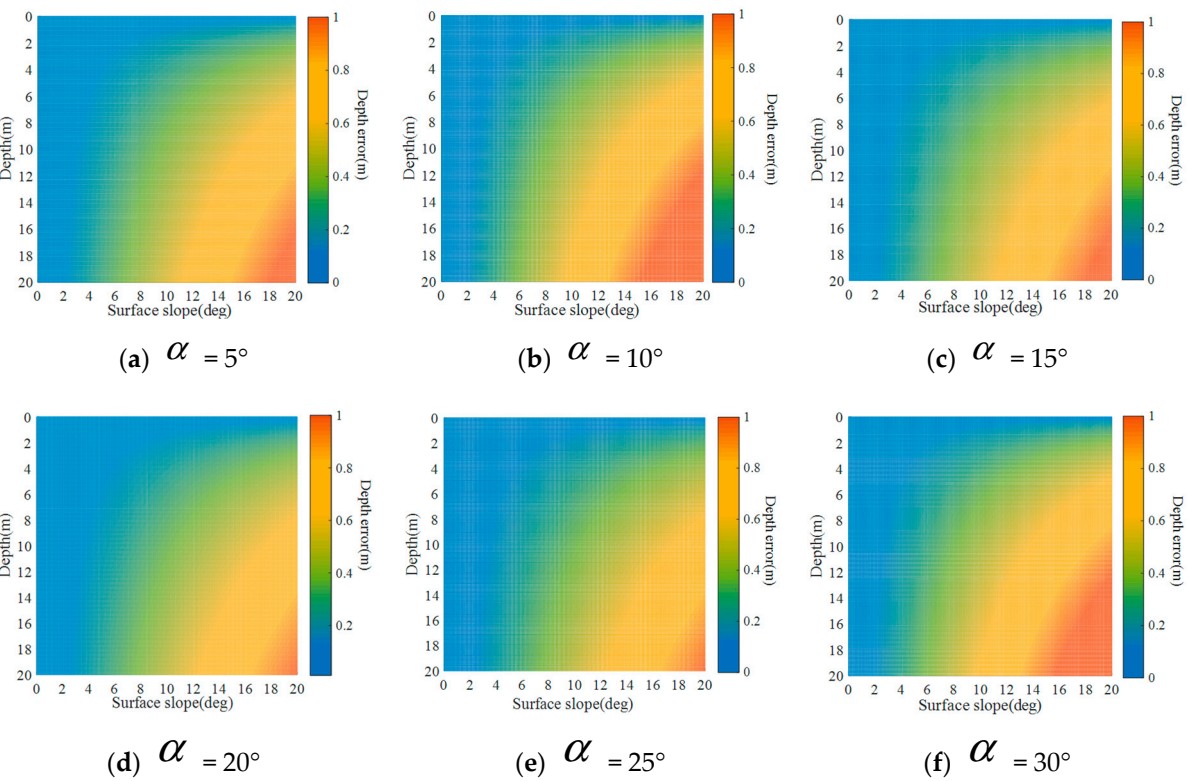

**Figure 23.** The influence of the laser emission angle and the water surface slope on the depth error.

As the water depth of the target water area is less than 20 m, Figure 24 shows the depth deviation when the laser emission angle of the ALB system is 20°. The curves in the figure represent the minimum accuracy requirements of the IHO S-44 Special Order and Order 1a/1b.

Table 12 shows the minimum bathymetry standards for the safety of navigation hydrographic surveys and the corresponding allowable sea surface slope without incident angle correction in different depth ranges for the ALB system bathymetric results. TVU is a quantity with all contributing vertical measurement uncertainties included, and $\sigma_{\alpha-max}$ is the maximum allowable surface slope.

As in the above analysis, first, choosing a stable water surface environment is generally beneficial for ALB water surface incidence correction. In addition, waves will adversely affect the applicability of the ALB system's depth measurement results. As shown in Table 12, when the surface slope is greater than 5.85° and 11.57°, the bathymetric error with a depth of more than 10 m may exceed the accuracy requirements proposed by the IHO S-44 Special Order and Order 1a/1b, respectively. In other words, if the bathymetric

error is larger than the corresponding order standards, the water surface waves will cause obvious interference to the ALB system's bathymetric accuracy; in that case, the adjustment of the ALB system's sounding results by the laser incidence correction model is necessary.

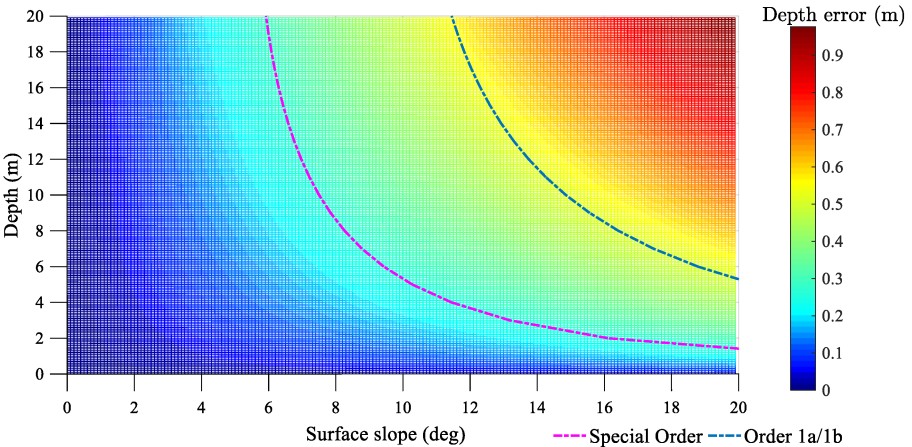

**Figure 24.** The relationship between sounding error vs. depth and surface inclination.

**Table 12.** The influence of sea surface inclination in different depth ranges on the sounding results.

| Depth (m) | Special Order | | Special Order 1a/1b | |
|---|---|---|---|---|
| | TVU (m) | $\sigma_{\alpha-\max}$ (deg) | TVU (m) | $\sigma_{\alpha-\max}$ (deg) |
| $\leq 5$ | $\leq 0.26$ | 15.50° | $\leq 0.51$ | 15.90° |
| $\leq 10$ | $\leq 0.27$ | 5.85° | $\leq 0.53$ | 11.57° |
| $\leq 15$ | $\leq 0.29$ | 5.05° | $\leq 0.55$ | 9.86° |
| $\leq 20$ | $\leq 0.31$ | 4.69° | $\leq 0.58$ | 9.03° |

## 5. Conclusions

Regarding the problem of water depth detection errors caused by the undulations of the ocean surface, we adopted the dimension-based self-adaptive laser incidence correction method after water surface point cloud preprocessing.

This method uses the local laser footprints at the incident position, extracts the dimensional information related to the surface shape and realizes self-adaptive selection for the normal vector estimation neighborhood radius based on the information entropy. Furthermore, we evaluated and analyzed the bathymetric performance of the method by simulation data experiments and actual measured experiments in the South China Sea and then obtained the following conclusions:

(1) **Accuracy.** The method suggested in this paper can not only effectively reflect the undulation characteristics of the sea surface microenvironment but also ensure the accuracy of the regional overall normal vector estimation in wave surfaces with different complexities. The adjustment result can meet the accuracy requirements of IHO S44.

(2) **Adaptability.** Since the proposed method does not involve waveform signal processing and LiDAR structure parameters, it can be widely used for a variety of ALB systems with different system scanning modes in laser incident correction on the air–water interface and underwater target detection adjustment.

(3) **Stability.** The self-adaptive method for the laser incidence correction method proposed in this paper has a stable effect for different wind speeds, seafloor materials and uneven footprint distributions without synchronous environmental parameter observations and additional presets during processing.

In addition, we analyzed the ALB system bathymetry errors caused by water surface waves and different laser emission angles based on the laser incidence correction method

proposed in this paper. The conditions for adopting the laser adaptive incident correction method were studied quantitatively. This provides a reference for the plan of ALB system operation and laser data processing.

In this paper, we analyzed the detection data from a small-footprint system, but the accuracy and effectiveness of the ALB system with a larger divergence angle (>1 mrad) should be further verified. As the statistical results of ocean slopes caused by wind waves were inversed for the method performance evaluation in the experiment, constructing a more refined inversion model would be helpful for the extraction of ocean surface dynamic information. Furthermore, the applicability of the ALB system in the nearshore area was confirmed in the experimental section, and the study of a system error compensation model combined with acoustic measurement is a direction that needs emphasis in future studies.

**Author Contributions:** K.G., Q.L., Q.M., C.W. and A.W. conceived and designed the experiments; K.G., J.Z., Y.L., D.Z. and W.X. contributed equally to analyzing the data; K.G. performed the experiments and wrote the paper. All authors have read and agreed to the published version of the manuscript.

**Funding:** This research was funded by the National Science Foundation of China, grant number 42001403.

**Institutional Review Board Statement:** Not applicable.

**Informed Consent Statement:** Not applicable.

**Acknowledgments:** The authors would like to thank the anonymous reviewers for their comments. Their insightful suggestions significantly improved this paper. This work was supported in part by the National Science Foundation of China (Grant No. 42001403), the National Natural Science Foundation of China (grant No. 41974006), Shenzhen Scientific Research and Development Funding programs (grant No. KQJSCX20180328093453763 and JCYJ20180305125101282) and the Department of Education of Guangdong Province (grant No. 2018KTSCX196).

**Conflicts of Interest:** The authors declare no conflict of interest.

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
