# Peer review of "Errors of Airborne Bathymetry LiDAR Detection Caused by Ocean Waves and Dimension-Based Laser Incidence Correction"

_remotesensing, doi:10.3390/rs13091750_

Round 1
Reviewer 1 Report
This study proposes a method that will help minimize the effects of ocean wave for accurate derivation of water depth with an ALB. It was further demonstrated. Given the importance of bathymetry mapping and more specifically, the role tools such as ALB plays (in e.g. in using bathymetry estimates from ALB for assessing the performance of satellite estimates based on physics based models, a study like this will be of benefit in ocean remote sensing communities.
While I recommend this article for publication, authors should further show the novelty of this study. It will be important to be very clear on how the authors' method is different from other methods/approaches that attempt to achieve similar thing using ALB.
Few other things to note:
Avoid the use of "in 2013 - e.g. line 66 - no need for such usage.
line 227 - how can we ascertain this significance? Any statistical test to show this?
line 462 - I don't think a new paragraph is needed here.
Author Response
Thank you very much for your comments and suggestions.We have studied comments carefully and have made correction which we hope meet with approval. Revised portion are marked in blue in the paper.

Reviewer 2 Report
The introduction provides sufficient background, however, it can be enriched by supplementing proper research objectives/hypothesis of this study, not only the challenges. After that, the responses for all research objectives should be addressed at the beginning of the discussion. Moreover, the introduction may be enriched for the potential utilization of shallow water bathymetry DEM. Please, consider enriching the introduction including some very recent works covering the following potential applications, e.g.:
coastal zone assessment:
- Tysiac, P. Bringing Bathymetry LiDAR to Coastal Zone Assessment: A Case Study in the Southern Baltic. Remote Sensing 2020, 12, doi:10.3390/rs12223740.
underwater archaeology:
- Janowski, L., Kubacka, M., Pydyn, A., Popek, M., Gajewski, L., 2021. From acoustics to underwater archaeology: Deep investigation of a shallow lake using high‐resolution hydroacoustics—The case of Lake Lednica, Poland. Archaeometry.
benthic habitat mapping:
- Fogarin, S.; Madricardo, F.; Zaggia, L.; Sigovini, M.; Montereale‐Gavazzi, G.; Kruss, A.; Lorenzetti, G.; Manfé, G.; Petrizzo, A.; Molinaroli, E.; Trincardi, F. Tidal inlets in the Anthropocene: geomorphology and benthic habitats of the Chioggia inlet, Venice Lagoon (Italy). Earth Surface Processes and Landforms 2019, 44, 2297–2315, doi:10.1002/esp.4642.
The discussion section requires clarification and supplementation for at least the parts listed below. While the authors describe their results rather properly, they rarely compared them to other works. There is a vague reference to the implications of this study and what the results mean to a wider scientific community. Moreover, the authors treated the limitations of the research very shortly and did not address recommendations for future research, e.g., possible calibration of your method performance with high-resolution bathymetry from multibeam echosounder.
Some specific comments:
line 469: refer to the technical details of iGreena listed in Table 3.
line 795: since you provided single-beam depth points, I encourage you to supplement them with measured depth bathymetry for each point.
Author Response

(The authors gave the same response as above.)

Round 2
Reviewer 1 Report
Thanks you for responding to my requests for finetuning your paper. The paper is ready to be published; however, I will encourage that authors should proofread/engage a professional editor to ensure readability, e.g.:
Lines 56 to 58: lack of clarity: topic sentence does not match with the content in the paragraph; use of 'and' twice seems not appropriate
use of 'just' in line 969.
use of 'has' in line 87.
unnecessary use of capitalization in line 109.
Line 142 - change 'the' to 'this'
Be consistent with your tenses - see lines 142 and 146
Familiarise yourself with the the referencing format of Remote Sensing." When you use et al, the reference number should come after this not at the end of the sentence.
Author Response
Dear Editors and Reviewers:
Thank you for your letter and for the comments concerning our manuscript entitled “Errors of Airborne Bathymetry LiDAR Detection Caused by Ocean Waves and Dimension-Based Laser Incidence Correction” (Manuscript ID: remotesensing-1177290). The comments are all very helpful for revising and improving our paper, as well as the important guiding significance to our research.
We have checked and polished the manuscript for the language and form and modified the manuscript using Microsoft Word Track Changes in the new uploaded file. The corrections in the paper and the responses to the reviewer’s comments are as follows:
Point 1: Lines 56 to 58: lack of clarity: topic sentence does not match with the content in the paragraph; use of 'and' twice seems not appropriate
Response 1: We appreciate the reviewer's comment, the sentence has been re-edited as follow:
Lines 59 to 60
However, the ocean is a complex physical system, the ocean surface is the interface between air and water which always maintains irregular undulations under the influence of currents, tides, winds and other dynamic factors.
Point 2: use of 'just' in line 969.
Response 2: Thanks to this reviewer’s comment, ‘just’ has been deleted in the sentence. (Line 975)
Point 3: use of 'has' in line 87.
Response 3: We deleted the ‘has’ in the sentence. (Line 90)
Lines 90 to 92
Li [25]conducted a statistical study on the water surface slope caused by wind and waves and summarized the changes in the ALB system detection errors caused by wave surface inclination under different wind speed conditions.
Point 4: unnecessary use of capitalization in line 109.
Response 4: Thanks for the comment, and we have made corrections in the line 112.
Point 5: Line 142 - change 'the' to 'this'
Response 5: the relevant content has been modified in Line 145
Point 6: Be consistent with your tenses - see lines 142 and 146
Response 6: The correction has been made and the revised text is as follow:
Lines 145 to 153
In this paper, we study the method for correcting the laser incidence angle on undulating ocean surfaces and apply a self-adaptive correction model based on the raw point cloud dimension to improve the applicability of ALB systems. The simulated undulation surface data and the measured data are used to verify and analyze the accuracy and practicability of the correction model. We use measured data with two scanning modes from distinct experimental areas in the South China Sea. On this basis, the effect of surface waves on the depth detection capability of an ALB system is analyzed with a quantitative approach, which provides some effective references for ALB data acquisition.
Point 7: Familiarize yourself with the referencing format of Remote Sensing." When you use et al, the reference number should come after this not at the end of the sentence.
Response 7: We have made correction according to the referencing format, and the modification could be referred to our latest uploaded file.
In addition, we have modified the Figure 2.in the line 172, that “Seafloor laser-footprints position” has been changed to “Seafloor laser-footprint position” in the flow chart.
We earnestly appreciate the Editors/Reviewers’ work and hope that the corrections could be met with approval.
Once again, thank you very much for your comments and suggestions.
Yours sincerely
